# NEURAL NETWORKS WITH LATE-PHASE WEIGHTS

**Johannes von Oswald\*, Seijin Kobayashi\*, João Sacramento\*,**
**Alexander Meulemans, Christian Henning, Benjamin F. Grewe**
\* – equal contribution

Institute of Neuroinformatics
University of Zürich and ETH Zürich
Zürich, Switzerland
`{voswaldj,seijink,rjoao,ameulema,henningc,bgrewe}@ethz.ch`

## ABSTRACT

The largely successful method of training neural networks is to learn their weights using some variant of stochastic gradient descent (SGD). Here, we show that the solutions found by SGD can be further improved by ensembling a subset of the weights in late stages of learning. At the end of learning, we obtain back a single model by taking a spatial average in weight space. To avoid incurring increased computational costs, we investigate a family of low-dimensional late-phase weight models which interact multiplicatively with the remaining parameters. Our results show that augmenting standard models with late-phase weights improves generalization in established benchmarks such as CIFAR-10/100, ImageNet and `enwik8`. These findings are complemented with a theoretical analysis of a noisy quadratic problem which provides a simplified picture of the late phases of neural network learning.

## 1 INTRODUCTION

Neural networks trained with SGD generalize remarkably well on a wide range of problems. A classic technique to further improve generalization is to ensemble many such models (Lakshminarayanan et al., 2017). At test time, the predictions made by each model are combined, usually through a simple average. Although largely successful, this technique is costly both during learning and inference. This has prompted the development of ensembling methods with reduced complexity, for example by collecting models along an optimization path generated by SGD (Huang et al., 2017), by performing interpolations in weight space (Garipov et al., 2018), or by tying a subset of the weights over the ensemble (Lee et al., 2015; Wen et al., 2020).

An alternative line of work explores the use of ensembles to guide the optimization of a single model (Zhang et al., 2015; Pittorino et al., 2020). We join these efforts and develop a method that fine-tunes the behavior of SGD using late-phase weights: late in training, we replicate a subset of the weights of a neural network and randomly initialize them in a small neighborhood. Together with the stochasticity inherent to SGD, this initialization encourages the late-phase weights to explore the loss landscape. As the late-phase weights explore, the shared weights accumulate gradients. After training we collapse this implicit ensemble into a single model by averaging in weight space.

Building upon recent work on ensembles with shared parameters (Wen et al., 2020) we explore a family of late-phase weight models involving multiplicative interactions (Jayakumar et al., 2020). We focus on low-dimensional late-phase models that can be ensembled with negligible overhead. Our experiments reveal that replicating the ubiquitous batch normalization layers (Ioffe & Szegedy, 2015) is a surprisingly simple and effective strategy for improving generalization[1]. Furthermore, we find that late-phase weights can be combined with stochastic weight averaging (Izmailov et al., 2018), a complementary method that has been shown to greatly improve generalization.

---

[1]We provide code to reproduce our experiments at `https://github.com/seijin-kobayashi/late-phase-weights`

## 2 METHODS AND MODELS

### 2.1 LEARNING WITH LATE-PHASE WEIGHTS

**Late-phase weights.** To apply our learning algorithm to a given neural network model $f_w$ we first specify its weights $w$ in terms of two components, base and late-phase ($\theta$ and $\phi$, resp.). The two components interact according to a weight interaction function $w = h(\theta, \phi)$. Base weights are learned throughout the entire training session, and until time step $T_0$ both $\theta$ and $\phi$ are learned and treated on equal grounds. At time step $T_0$, a hyperparameter of our algorithm, we introduce $K$ late-phase components $\Phi = \{\phi_k\}_{k=1}^K$, that are learned together with $\theta$ until the end.

This procedure yields a late-phase ensemble of $K$ neural networks with parameter sharing: reusing the base weights $\theta$, each late-phase weight $\phi_k$ defines a model with parameters $w_k = h(\theta, \phi_k)$.

**Late-phase weight averaging at test time.** Our ensemble defined by the $K$ late-phase weight configurations in $\Phi$ is kept only during learning. At test time, we discard the ensemble and obtain a *single* model by averaging over the $K$ late-phase weight components. That is, given some input pattern $x$, we generate a prediction $\overline{y}(x)$ using the averaged model, computed once after learning:

$$\overline{y}(x) = f_{\overline{w}}(x), \qquad \overline{w} \equiv h\left(\theta, \frac{1}{K}\sum_{k=1}^K \phi_k\right). \tag{1}$$

Hence, the complexity of inference is independent of $K$, and equivalent to that of the original model.

**Late-phase weight initialization.** We initialize our late-phase weights from a reference base weight. We first learn a base parameter $\phi_0$ from time step $t = 0$ until $T_0$, treating $\phi_0$ as any other base parameter in $\theta$. Then, at time $t = T_0$, each configuration $\phi_k$ is initialized in the vicinity of $\phi_0$. We explore perturbing $\phi_0$ using a symmetric Gaussian noise model,

$$\phi_k = \phi_0 + \frac{\sigma_0}{Z(\phi_0)}\epsilon_k, \tag{2}$$

where $\epsilon_k$ is a standard normal variate of appropriate dimension and $\sigma_0$ is a hyperparameter controlling the noise amplitude. We allow for a $\phi_0$-dependent normalization factor, which we set so as to ensure layerwise scale-invariance, which helps finding a single $\sigma_0$ that governs the initialization of the entire network. More concretely, for a given neural network layer $l$ with weights $\phi_0^{(l)}$ of dimension $D^{(l)}$, we choose $Z(\phi_0^{(l)}) = \sqrt{D^{(l)}}/\|\phi_0^{(l)}\|$.

Our perturbative initialization (Eq. 2) is motivated by ongoing studies of the nonconvex, high-dimensional loss functions that arise in deep learning. Empirical results and theoretical analyses of simplified models point to the existence of dense clusters of connected solutions with a locally-flat geometry (Hochreiter & Schmidhuber, 1997a) that are accessible by SGD (Huang et al., 2017; Garipov et al., 2018; Baldassi et al., 2020). Indeed, the eigenspectrum of the loss Hessian evaluated at weight configurations found by SGD reveals a large number of directions of low curvature (Keskar et al., 2017; Chaudhari et al., 2019; Sagun et al., 2018). For not yet completely understood reasons, this appears to be a recurring phenomenon in overparameterized nonlinear problems (Brown & Sethna, 2003; Waterfall et al., 2006).

Based on these observations, we assume that the initial parameter configuration $\phi_0$ can be perturbed in a late phase of learning without leading to mode hopping across the different models $w_k$. While mode coverage is usually a sought after property when learning neural network ensembles (Fort et al., 2020), here it would preclude us from taking the averaged model at the end of learning (Eq. 1).

**Stochastic learning algorithm.** Having decomposed our weights into base and late-phase components, we now present a stochastic algorithm which learns both $\theta$ and $\Phi$. Our algorithm works on the standard stochastic (minibatch) neural network optimization setting (Bottou, 2010). Given a loss function $\mathcal{L}(\mathcal{D}, w) = \frac{1}{|\mathcal{D}|}\sum_{x\in\mathcal{D}} L(x, w)$ to be minimized with respect to the weights $w$ on a set of data $\mathcal{D}$, at every round we randomly sample a subset $\mathcal{M}$ from $\mathcal{D}$ and optimize instead the stochastic loss $\mathcal{L}(\mathcal{M}, w)$. However, in contrast to the standard setting, in late stages of learning ($t > T_0$) we simultaneously optimize $K$ parameterizations $\mathcal{W} := \{w_k \mid w_k = h(\theta, \phi_k)\}_{k=1}^K$, instead of one.

We proceed by iteration over $\mathcal{W}$. At each step $k$, we sample a minibatch $\mathcal{M}_k$ and immediately update the late-phase weights $\phi_k$, while accumulating gradients over the shared base weights $\theta$. Such gradient accumulation has been previously used when learning ensembles (Lee et al., 2015; Wen et al., 2020) and multi-task models (Rebuffi et al., 2017) with shared base parameters. A single iteration is finally concluded by changing the base weights in the direction opposite of the accumulated gradient. We scale the accumulated gradient by $\gamma_\theta$; setting $\gamma_\theta = 1/K$ recovers the original step size in $\theta$, but other choices are possible. In particular, we find that a large $\gamma_\theta$ of unit size is in practice often tolerated, resulting in accelerated learning.

We summarize an iteration of our method in Algorithm 1, where the loss $\mathcal{L}(\mathcal{M}, \theta, \phi)$ is now seen as a function of $\theta$ and $\phi$. We opt for a general presentation using unspecified gradient-based update operators $U_\phi$ and $U_\theta$. These operators can be set to optimizers of choice. For instance, our method might benefit from additional noise injection onto parameter updates (Welling & Teh, 2011). Furthermore, late-phase optimizers need not coincide with the optimizer used in the early phase. In our work we typically set $U_\phi$ and $U_\theta$ to a single step of SGD with Nesterov momentum (Nesterov, 2004), and explore Adam (Kingma & Ba, 2015) and plain SGD in a smaller set of experiments.

---

**Algorithm 1:** Late-phase learning

**Require:** Base weights $\theta$,
late-phase weight set $\Phi$, dataset $\mathcal{D}$,
gradient scale factor $\gamma_\theta$, loss $\mathcal{L}$
**Require:** Training iteration $t > T_0$
**for** $1 \le k \le K$ **do**
$\quad \mathcal{M}_k \leftarrow$ Sample minibatch from $\mathcal{D}$
$\quad \Delta\theta_k \leftarrow \nabla_\theta \mathcal{L}(\mathcal{M}_k, \theta, \phi_k)$
$\quad \phi_k \leftarrow U_\phi(\phi_k, \nabla_{\phi_k} \mathcal{L}(\mathcal{M}_k, \theta, \phi_k))$
$\theta \leftarrow U_\theta(\theta, \gamma_\theta \sum_{k=1}^{K} \Delta\theta_k)$

---

## 2.2 LATE-PHASE WEIGHT MODELS

As detailed next, we consider a number of distinct late-phase weight models in our experiments. In particular, we explore weight interaction functions $h$ in which late-phase weights have low dimensionality, to avoid a large increase in complexity with the ensemble size $K$. To counteract this reduced dimensionality, we make extensive use of multiplicative base-late weight interactions. This design choice is motivated by the large expressive power of multiplicative interactions despite low dimensionality, which has been demonstrated in a wide range of settings (Jayakumar et al., 2020).

**Late-phase batch normalization layers.** Batch normalization layers (BatchNorm; Ioffe & Szegedy, 2015) are a staple of current deep neural network models. Besides standardizing the activity of the layer they are applied to, BatchNorm units introduce a learnable multiplicative (scale) parameter $\gamma$ and an additive (shift) parameter $\beta$. While being low-dimensional, these additional parameters have large expressive power: it has been shown that learning only $\gamma$ and $\beta$ keeping the remaining weights frozen can lead to significantly lower loss than when learning random subsets of other weights of matching dimensionality (Frankle et al., 2020; Mudrakarta et al., 2019).

We take the scale and shift parameters of BatchNorm layers as our first choice of late-phase weights; the base weights are the remaining parameters of the model. Batch statistics are also individually estimated for each model in $\mathcal{W}$. This late-phase weight parameterization is motivated by (*i*) the expressive power of $\gamma$ and $\beta$ discussed above, and by (*ii*) practical considerations, as BatchNorm layers are generally already present in feedforward neural network models, and are otherwise easy to implement efficiently.

More concretely, let us consider an affine transformation layer $l$ which maps an input vector $r^{(l-1)}$ to $\theta_w^{(l)} r^{(l-1)} + \theta_b^{(l)}$, where the early-phase weight matrix $\theta_w^{(l)}$ and bias vector $\theta_b^{(l)}$ are already standardized using the respective batch statistics. For this standard layer, our model introduces a multiplicative interaction between base and late-phase weights, $\mathrm{diag}(\gamma^{(l)}) \theta_w^{(l)}$, and an additive interaction between base and late-phase bias parameters, $\theta_b^{(l)} + \beta^{(l)}$.

**Late-phase rank-1 matrix weights.** We also study a closely related late-phase weight model, where existing weight matrices – the base components, as before – are multiplied elementwise by rank-1 matrices (Wen et al., 2020). For a given affine layer $l$, we define a late-phase weight matrix with resort to a pair of learnable vectors, $\phi^{(l)} = u^{(l)} v^{(l)^T}$. Taking the Hadamard product with the base weight matrix yields the effective weights $W^{(l)} = \phi^{(l)} \circ \theta^{(l)}$.

With this parameterization, we recover the ensemble proposed by Wen et al. (2020), except that here it is generated late in training using our perturbative initialization (Eq. 2). Unlike BatchNorm layers, which include the shift parameter, rank-1 late-phase weights interact in a purely multiplicative manner with base weights. We study this model since it is easy to implement on neural networks which do not feature BatchNorm layers, such as standard long short-term memories (LSTMs; Hochreiter & Schmidhuber, 1997b).

**Hypernetworks with late-phase weight embeddings.** Additionally, we generalize the late-phase weight models described above using hypernetworks (Ha et al., 2017). A hypernetwork generates the parameters $w$ of a given target neural network $f_w$ based on a weight embedding. In our framework, we can use a hypernetwork to implement the interaction function $w = h(\theta, \phi)$ directly, with parameters $\theta$ corresponding to base weights and embeddings $\phi$ to late-phase weights.

We experiment with linear hypernetworks and use the same hypernetwork to produce the weights of multiple layers, following Savarese & Maire (2019); Ha et al. (2017); von Oswald et al. (2020). In this scheme, the weight embedding input specifies the target layer whose parameters are being generated. More specifically, the weight matrix for some layer $l$ belonging to a group of layers $g$ which share a hypernetwork is given by $W^{(g,l)} = \theta^{(g)} \phi^{(g,l)}$, where $\theta^{(g)}$ and $\phi^{(g,l)}$ are appropriately-sized tensors. Sharing $\theta^{(g)}$ over a layer group $g$ allows countering an increase in the overall number of parameters. We parameterize our hypernetworks such that the weight embedding vectors $\phi^{(g,l)}$ are small, and therefore cheap to ensemble.

**Late-phase classification layers.** Finally, inspired by Lee et al. (2015), in classification experiments we take the weights of the last linear layer as late-phase weights by default. In modern neural network architectures these layers do not usually comprise large numbers of parameters, and our architecture explorations indicated that it is typically beneficial to ensemble them. We therefore include $W^{(L)}$ in our late-phase weights $\phi$, where $W^{(L)}$ denotes the weights of the final layer $L$.

## 3 RESULTS

### 3.1 NOISY QUADRATIC PROBLEM ANALYSIS

Before turning to real-world learning problems, we first focus on a simplified stochastic optimization setup which can be analytically studied. We consider the noisy quadratic problem (NQP; Schaul et al., 2013; Martens, 2016; Wu et al., 2018; Zhang et al., 2019a;b), where the goal is to minimize the scalar loss

$$\mathcal{L} = \frac{1}{2}(w - w^* + \epsilon)^T H (w - w^* + \epsilon) \tag{3}$$

with respect to $w \in \mathbb{R}^n$. In the equation above, $w^*$ denotes the target weight vector, which is randomly shifted by a noise variable $\epsilon$ assumed to follow a Gaussian distribution $\mathcal{N}(0, \Sigma)$. The (constant) Hessian matrix $H$ controls the curvature of the problem.

Despite the simplicity of Eq. 3, the NQP captures a surprising number of empirically-observed aspects of neural network learning (Zhang et al., 2019a). Here, we motivate its study as a model of late stages of learning, by Taylor expanding the loss around a minimum $w^*$. Thus, for a sufficiently late initialization time $T_0$ (and small $\sigma_0$) the NQP is particularly well suited to study our algorithm.

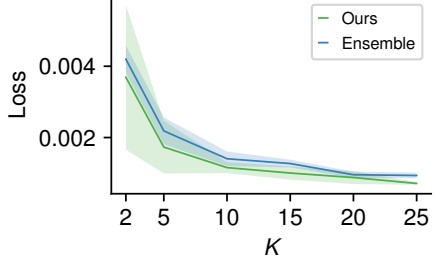

Figure 1: Steady-state loss for varying $K$, of multiplicative late-phase weights (*Ours*) compared to an ensemble of models.

There are three main strategies to improve the expected NQP loss after convergence: (*i*) increase the minibatch size $B$, (*ii*) use more members $K$ in an ensemble, and (*iii*) decrease the learning rate $\eta$ (Zhang et al., 2019a). Our Algorithm 1 combines the first two strategies in a non-trivial manner. First, the gradients for base weights $\theta$ are averaged during the inner loop over all ensemble members, corresponding to a minibatch-size rescaling by $K$. Second, we

introduce $K$ ensemble members, to be averaged in weight space, that only differ in their late-phase weights $\phi$.

In Appendix C, we show analytically that this combination of an increased effective minibatch size for $\theta$ and introducing $K$ ensemble members for $\phi$ is successful, resulting in a scaling of the expected loss after convergence by $\frac{1}{K}$. This analysis holds for general $\Sigma$ and $H$, and for both scalar and hypernetwork multiplicative late-phase weights. Hence, our approach combines the benefits of an increased effective minibatch size and of ensembling, while yielding a single model after training.

We present a numerical validation of this theoretical result in Fig. 1. Our model includes a multiplicative late-phase weight, $w_k = \theta\,\phi_k$ with $\phi_k \in \mathbb{R}$ and $\theta \in \mathbb{R}^n$. We simulate a standard instance of the NQP, with diagonal Hessian $H_{ii} = 1/i$ and $\Sigma = H^{-1}$ (cf. Zhang et al., 2019a), and report the average loss after convergence. Hyperparameters are given in Appendix C. As predicted by the theory, the loss falls as $\sim 1/K$ with increasing ensemble size $K$, and our algorithm performs on par with a full ensemble of $K$ models trained independently with gradient descent.

## 3.2 CIFAR-10/100 EXPERIMENTS

To test the applicability of our method to more realistic problems, we next augment standard neural network models with late-phase weights and examine their performance on the CIFAR-10 and CIFAR-100 image classification benchmarks (Krizhevsky, 2009). We use standard data preprocessing methods (cf. Appendix A) and train our models for 200 epochs from random initializations, except when noted otherwise. All evaluated methods are trained using the same amount of data.

Besides SGD (with Nesterov momentum), we also investigate stochastic weight averaging (SWA; Izmailov et al., 2018), a recent reincarnation of Polyak averaging (Polyak & Juditsky, 1992) that can strongly improve neural network generalization. For completeness, we present pseudocode for SWA in Algorithm 2 and SGD with Nesterov momentum in Algorithm 3 (cf. Appendix A). When learning neural networks with late-phase weights we set $U_\phi$ and $U_\theta$ to one step of SGD (or SGD wrapped inside SWA).

We compare our method to dropout (Srivastava et al., 2014), a popular regularization method that can improve generalization in neural networks. Like our approach, dropout produces a single model at the end of training. We also consider its Monte Carlo variant (MC-dropout; Gal & Ghahramani, 2016), and the recently proposed BatchEnsemble (Wen et al., 2020). This method generates an ensemble using rank-1 matrices as described in Section 2.2. Predictions still need to be averaged over multiple models, but this averaging step can be parallelized in modern hardware.

Additionally, we report single-seed results obtained with an ensemble of $K$ independently-trained models (a deep ensemble, Lakshminarayanan et al., 2017). Deep ensembles provide a strong baseline, at the expense of large computational and memory costs. Therefore, they are not directly comparable to the other methods considered here, and serve the purpose of an upper baseline.

By contrast, augmenting the architectures considered here with late-phase weights results in negligible additional costs during learning (with the exception of hypernetworks, which require additional tensor products) and none during testing. In principle, a set of independently-trained models yielded by our algorithm can therefore even be used as the basis of a deep ensemble, when the memory and compute budget allows for one. We present proof-of-concept experiments exploring this option.

Throughout our CIFAR-10/100 experiments we set $K = 10$, use a fast base gradient scale factor of $\gamma_\theta = 1$, and set our late-phase initialization hyperparameters to $T_0 = 120$ (measured henceforth in epochs; $T_0 = 100$ for SWA) and do not use initialization noise, $\sigma_0 = 0$. These hyperparameters were tuned manually once on CIFAR-100 and then kept fixed unless otherwise

Table 1: CIFAR-10, WRN 28-10. Mean $\pm$ std. over 5 seeds. Late-phase BatchNorm (LPBN).

| Model | Test acc. (%) |
|---|---|
| Base (SGD) | $96.16^{\pm 0.12}$ |
| Dropout (SGD) | $96.02^{\pm 0.06}$ |
| MC-Dropout (SGD) | $96.03^{\pm 0.09}$ |
| BatchEnsemble (SGD) | $96.19^{\pm 0.18}$ |
| Late-phase (SGD) | $\mathbf{96.46}^{\pm 0.15}$ |
| Base (SWA) | $96.48^{\pm 0.04}$ |
| Late-phase (SWA) | $\mathbf{96.81}^{\pm 0.07}$ |
| Deep ensemble (SGD) | $96.91$ |
| Deep ensemble (LPBN, SGD) | $\mathbf{96.99}$ |

noted. We use standard learning rate scheduling, optimized for SGD and SWA on the base model (cf. Appendices A and B). Last-layer weights are included by default in our late-phase weight set $\Phi$.

**CIFAR-10.** For CIFAR-10 we focus on the WRN architecture, a high-performance residual network (WRN; Zagoruyko & Komodakis, 2016) which features BatchNorm layers. Taking advantage of this we implement a late-phase weight model consisting of BatchNorm shift and scale parameters.

All algorithms achieve a training error close to zero (cf. Appendix B). The resulting predictive accuracies are shown in Table 1. We find that augmenting the WRN 28-10 (a standard WRN configuration) with BatchNorm late-phase weights leads to a systematic improvement in generalization, reducing the gap with a deep ensemble of $K = 10$ models. Initializing our ensemble from the onset ($T_0 = 0$) fails to meet the performance of the base model, reaching only $95.68 \pm 0.23\%$ (cf. Appendix 12).

We also investigate initializing a late-phase (full) deep ensemble at $T_0 = 120$. This results in a test set accuracy of $96.32 \pm 0.09\%$, in between late-phase BatchNorm weights and no late-phase weights at all. This speaks to the data-efficiency of our low-dimensional late-phase ensembles which can be trained with as little data as a single model, besides being memory efficient.

In addition, we consider a larger instance of the WRN model (the WRN 28-14), trained for 300 epochs using cutout data augmentation (DeVries & Taylor, 2017), as well as a small convolution neural network without skip connections, cf. Table 3. When late-phase weights are employed in combination with SWA, we observe significant accuracy gains on the WRN 28-14. Thus, our late-phase weights impose an implicit regularization that is effective on models with many weights. Similarly, we observe larger gains when training on a random subset of CIFAR-10 with only $10^4$ examples (cf. Appendix B).

Table 2: Mean CIFAR-100 test set accuracy (%) $\pm$ std. over 5 seeds, WRN 28-10. Different late-phase weight augmentations are compared to the base architecture and to an upper bound consisting of an ensemble of models. *Deep ens.* stands for deep ensemble, *LPBN* for late-phase BatchNorm.

|  | Base | BatchNorm | Hypernetwork | Deep ens. | Deep ens. (LPBN) |
|---|---|---|---|---|---|
| SGD | $81.35^{\pm 0.16}$ | $82.87^{\pm 0.22}$ | $81.55^{\pm 0.31}$ | 84.09 | 84.69 |
| SWA | $82.46^{\pm 0.09}$ | $83.06^{\pm 0.08}$ | $82.01^{\pm 0.17}$ | 83.62 | - |

**CIFAR-100.** We next turn to the CIFAR-100 dataset, which has 10-fold less examples per class and more room for improvements. We study the WRN 28-10, as well as the larger WRN 28-14 variant (using cutout data augmentation as before) and a PyramidNet (Han et al., 2017) with ShakeDrop regularization (Yamada et al., 2019). The latter are trained for 300 epochs.

Predictive accuracy is again highest for our neural networks with late-phase weights, trained with SGD or SWA, cf. Table 2. We observe that the simplest BatchNorm late-phase weight model reaches the highest accuracy, with late-phase hypernetwork weight embeddings yielding essentially no improvements. Once again, the setting of $T_0 = 0$ (onset ensemble learning) fails to match base model performance, finishing at $80.26 \pm 0.42\%$ test accuracy. As for CIFAR-10, a late-phase full deep ensemble only reached intermediate improvements, at $82.17 \pm 0.15\%$ test accuracy. Furthermore, a gap towards deep ensembles persists. This suggests that covering different modes of the loss (Fort et al., 2020) can provide benefits that cannot be captured by ensembling models in a small neighborhood.

The final averaged solutions found with late-phase weights are strong base models to build a deep ensemble of independently-trained networks. The fact that our algorithm yields a single model allows further pushing the upper bound of what can be achieved when unrestricted full ensemble training is possible. This improvement comes at no cost compared to a standard deep ensemble.

Table 3: Additional architectures, CIFAR-10 (C10) and CIFAR-100 (C100). Mean test set acc. $\pm$ std. over 3 seeds (%). Late-phase BatchNorm weights.

|  | Base | Late-phase |
|---|---|---|
| C10 ConvNet (SGD) | $77.41^{\pm 0.23}$ | $77.94^{\pm 0.37}$ |
| C10 WRN 28-14 (SWA) | $96.75^{\pm 0.05}$ | $97.45^{\pm 0.10}$ |
| C100 WRN 28-14 (SWA) | $84.01^{\pm 0.29}$ | $85.00^{\pm 0.25}$ |
| C100 PyramidNet (SGD) | $84.04^{\pm 0.28}$ | $84.35^{\pm 0.14}$ |

We train additional neural network architectures restricting our experiments to the BatchNorm late-phase weight model, which can be readily implemented without architectural modifications. Again, learning with late-phase weights yields a consistent improvement over the baseline, cf. Table 3.

Notably, SWA can achieve high predictive accuracy with a large constant learning rate (Izmailov et al., 2018). We reproduce these previous results and show that they improve when learning with late-phase weights, cf. Fig. 2. Substantial progress is made both when entering the late-phase learning period and when activating SWA.

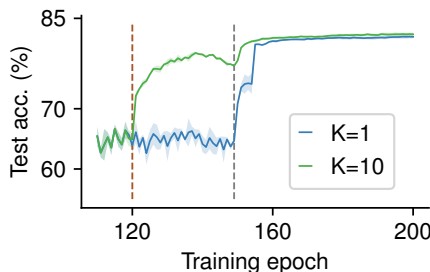

Figure 2: WRN 28-10, CIFAR-100, constant learning rate SWA (activated at epoch 150). With BatchNorm late-phase weights ($K$=10, initialized at epoch 120) and without ($K$=1). Mean test acc. (%) $\pm$ std. over 5 seeds.

**Out-of-distribution (OOD) generalization.** Deep ensembles are an effective technique for improving the behavior of neural networks in OOD data (Lakshminarayanan et al., 2017). We ask whether our implicit ensembles modeled during late-phase learning could confer a similar advantage to our final averaged model.

Additionally, we evaluate the performance of a late-phase weight ensemble obtained with large initialization noise $\sigma_0 = 0.5$ (at $T_0 = 100$), skipping the final weight averaging step. This requires integrating predictions over $K$ late-phase ensemble members at test time, $y(x) = \frac{1}{K} \sum_{k=1}^{K} y(x, w_k)$. Unlike standard deep ensembles, training this ensemble is still as cheap as training a single model.

Table 4: CIFAR-100, WRN-28-10, uncertainty representation results. Mean $\pm$ std. over 5 seeds (except deep ensembles). This first group of methods yield a single model; the second group requires test-time averaging over models while training as efficiently as $K$=1; the last group are full deep ensembles which require training $K$=10 models from scratch (*Deep ens.*). We report in-distribution test set acc. (%) and negative log-likelihood (NLL), and in-distribution vs. out-of-distribution (OOD) discrimination performance (average AUROC over four OOD datasets, see main text).

| | Test acc. (%) | Test NLL | OOD |
|---|---|---|---|
| Base (SGD) | $81.35^{\pm 0.16}$ | $0.7400^{\pm 0.0034}$ | $0.8015^{\pm 0.0189}$ |
| Dropout (Mean) (SGD) | $81.31^{\pm 0.20}$ | $0.7736^{\pm 0.0025}$ | $0.8022^{\pm 0.0299}$ |
| Late-phase Hypernetwork (SGD) | $81.55^{\pm 0.32}$ | $0.8327^{\pm 0.0066}$ | $0.8209^{\pm 0.00168}$ |
| Late-phase BatchNorm (SGD) | $\mathbf{82.87}^{\pm 0.14}$ | $\mathbf{0.7542}^{\pm 0.0076}$ | $\mathbf{0.8360}^{\pm 0.0118}$ |
| MC-Dropout (SGD) | $81.55^{\pm 0.11}$ | $0.7105^{\pm 0.0026}$ | $0.8225^{\pm 0.0488}$ |
| SWAG (SWA) | $82.12^{\pm 0.03}$ | $\mathbf{0.6189}^{\pm 0.0036}$ | $0.8283^{\pm 0.0274}$ |
| BatchEnsemble (SGD) | $81.25^{\pm 0.10}$ | $0.7691^{\pm 0.0048}$ | $0.8285^{\pm 0.0189}$ |
| Late-phase BatchNorm (SGD, non-averaged) | $\mathbf{82.71}^{\pm 0.10}$ | $0.7512^{\pm 0.0069}$ | $\mathbf{0.8624}^{\pm 0.0094}$ |
| Deep ens. (SGD) | 84.09 | $\mathbf{0.5942}$ | 0.8312 |
| Deep ens. (Late-phase BatchNorm, SGD) | $\mathbf{84.69}$ | 0.6712 | $\mathbf{0.8575}$ |

We draw novel images from a collection of datasets (SVHN, Netzer et al. (2011); LSUN, Yu et al. (2015); Tiny ImageNet; CIFAR-10) and present them to a WRN 28-10 trained on CIFAR-100. We use Shannon's entropy (Cover & Thomas, 2006) to measure the uncertainty in the output predictive distribution, which should be high for OOD and low for CIFAR-100 data. Overall performance is summarized using the area under the receiver operating characteristics curve (AUROC), averaged over all datasets. We report per-dataset results in Appendix B (Table 16) alongside experiments measuring robustness to corruptions in the input data (Hendrycks & Dietterich, 2019).

We compare our results to alternative methods with strong uncertainty representation: MC-dropout (Gal & Ghahramani, 2016), SWA-Gaussian (SWAG; Maddox et al., 2019) and BatchEnsemble (Wen et al., 2020). All three methods require integrating predictions over an ensemble at test time.

We find that learning with late-phase weights increases prediction uncertainty in OOD data, allowing for a significantly better separation between in and out-of-distribution examples, cf. Table 4. The

OOD performance of late-phase BatchNorm weights compares favorably to the alternative methods including deep ensembles, even when using a single weight-averaged model, while maintaining high predictive accuracy. Remarkably, keeping the late-phase BatchNorm ensemble at test time allows reaching the highest OOD performance throughout. Paired with non-zero initialization noise $\sigma_0 > 0$ (cf. Appendix B), this method results in the best OOD performance.

Despite our improved performance on both predictive accuracy (with late-phase BatchNorm) and OOD discrimination (with late-phase BatchNorm and hypernetwork embeddings), the test set negative log-likelihood (NLL; often used to assess predictive uncertainty, Guo et al., 2017) is surprisingly slightly worse for our solutions. This is aligned with the finding that SWA does not always significantly reduce NLL, even though predictive accuracy increases (Maddox et al., 2019).

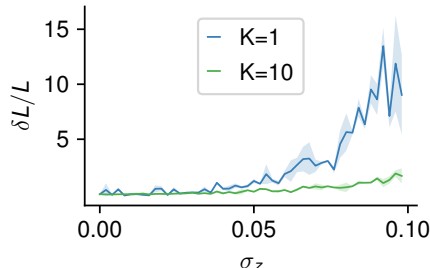

Figure 3: Flatness score. Mean score $\pm$ std. over 5 seeds, WRN 28-10, CIFAR-100, SGD, with and without Batch-Norm late-phase weights. Slower increase with $\sigma_z$ is better.

**Flatness.** Why do our networks generalize better? Approximate Bayesian inference suggests that flat minima generalize better than sharp minima (Hochreiter & Schmidhuber, 1997a; MacKay, 1992). Due to symmetries that are present in neural networks there is some debate surrounding this argument (Dinh et al., 2017), but current evidence seems favorable (Jiang et al., 2020).

We hypothesize that sharing base weights over $K$ late-phase weight configurations can implicitly lead to flatter solutions. To investigate whether our algorithm finds flatter minima, we examine a simple flatness score that correlates well with generalization (Pittorino et al., 2020; Jiang et al., 2020). Concretely, we add multiplicative Gaussian noise $z_i \sim \mathcal{N}(0, w_i^2 \sigma_z^2)$ to each weight $w_i$ and then measure the change in the loss $\delta\mathcal{L} = \mathbb{E}_z[\mathcal{L}(w + z) - \mathcal{L}(w)]$. Our final weight configurations are indeed in flatter regions of weight space according to this measure: $\delta L$ increases more slowly with $\sigma_z$ for the WRN 28-10 models that are learned with BatchNorm late-phase weights, Fig. 3.

### 3.3 IMAGENET EXPERIMENTS

To investigate whether our gains translate to large-scale learning problems, we train deep residual networks (He et al., 2016) and a densely-connected convolutional network (DenseNet; Huang et al., 2018) on the ImageNet dataset (Russakovsky et al., 2015). We start from pretrained models and contrast BatchNorm late-phase weight learning to fine-tuning with SGD for 20 epochs, with $\gamma_\theta = 1/K$

Table 5: Validation set acc. (%) on ImageNet. Mean $\pm$ std. over 5 seeds. BatchNorm late-phase and baseline trained for 20 epochs with SGD.

|  | Initial | Base | Late-phase |
|---|---|---|---|
| ResNet-50 | 76.15 | $76.62^{\pm0.06}$ | $76.87^{\pm0.03}$ |
| ResNet-152 | 78.31 | $78.37^{\pm0.01}$ | $78.77^{\pm0.01}$ |
| DenseNet-161 | 77.65 | $78.17^{\pm0.01}$ | $78.31^{\pm0.01}$ |

and $\sigma_0 = 0$ (cf. Appendix A). For simplicity we do not include last-layer weights in $\Phi$.

Fine-tuning with late-phase weights improves the final top-1 validation accuracy of this pretrained model significantly with only minor training, as seen in Table 5. These results serve as a proof-of-concept that existing models can be further improved, taking our late-phase initialization $T_0$ as the time the previous experimenter stopped training. In Appendix B, we present additional CIFAR-100 experiments where we apply late-phase learning starting at the suboptimal end-of-training $T_0 = 200$, to mimic the pretrained condition.

### 3.4 LSTM LANGUAGE MODELING EXPERIMENTS

Finally, we conduct experiments on the language modeling benchmark `enwik8`. To show that the benefits of late-phase weights extend to recurrent neural networks, we augment a standard LSTM with multiplicative late-phase weights consisting of rank-1 matrices (Wen et al., 2020, cf. Section 2.2).

Overfitting is a major issue when training LSTMs. Recent studies have shown that by leveraging vast amounts of computation and smart black-box optimizers (Golovin et al., 2017), properly regularized LSTMs can outperform previously published state-of-the-art models (Melis et al., 2017). To avoid this issue, we train models where the number of parameters ($\sim$1.56M) is drastically smaller than the number of training data points (90M), such that we

Table 6: `enwik8` results measured in bits per character (BPC), LSTM with 500 units. Mean over 5 seeds, with std. $\sigma < 0.01$ for all results.

| Model | Train | Test | Test (SWA) |
|---|---|---|---|
| Base | 1.570 | 1.695 | 1.626 |
| Base + Rank1 | 1.524 | 1.663 | 1.616 |
| Late-phase Rank1 | 1.522 | 1.633 | 1.615 |

do not observe any overfitting. Thus, we do not apply any regularization. This helps minimize the effects of hyperparameter tuning. Our only hyperparameter is the learning rate (0.001 here), which we tune via grid search to maximize base model performance.

We train our LSTM with 500 units for 50 epochs, optimizing every weight with Adam (Kingma & Ba, 2015). We apply a multiplicative rank-1 matrix elementwise to the recurrent weight matrix. Interestingly, merely adding the multiplicative parameters to the LSTM (Base) accelerates training and leads to better training and test set performance (measured in bits per character, BPC) with no additional changes to the optimizer (Base + Rank1, Table 6). Further improvements can be achieved with our late-phase weights. We generate $K = 10$ late-phase weight components at epoch 30 with $\sigma_0 = 0.35$ and set $\gamma_\theta = 1$. Additionally, we find that SWA (starting at epoch 40) substantially improves all scores, with smaller gains on the models with multiplicative weights.

## 4 RELATED WORK

Our late-phase weights define an ensemble with the special property that every model shares the same base weights. Such parameter sharing is an established method for ensembling neural networks while controlling for the memory and time complexity of learning (Lee et al., 2015). In designing our late-phase weight models, we draw directly from recent work which proposes sharing a set of base parameters over $K$ rank-1 matrices (Wen et al., 2020) or $K$ heads (Lee et al., 2015).

The elastic averaging SGD algorithm learns $K$ neural networks in parallel, coupled through an additional central model (EASGD; Zhang et al., 2015). Like our algorithm, EASGD often yields solutions which generalize better than those found by standard SGD (Pittorino et al., 2020). Our late-phase weight learning is intimately related to EASGD, as we optimize the performance of a central model through an ensemble. However, thanks to parameter sharing and late-phase ensembling, we do not find the need to introduce a coupling term to our loss function. Additionally, as we replicate a small number of parameters only, the complexity of our algorithm is greatly reduced in comparison to EASGD, which requires learning a full ensemble of models.

Splitting the weights of a neural network into a set of fast and slow components which vary on different timescales is a classic technique (Hinton & Plaut, 1987; Schmidhuber, 1992) that has proven useful in a wide range of problems. This list includes applications to few-shot learning (Munkhdalai & Yu, 2017; Nichol et al., 2018; Perez et al., 2018; Zintgraf et al., 2019; Flennerhag et al., 2020), optimization (Zhang et al., 2019b; Chaudhari et al., 2019), improving recurrent neural networks (Ba et al., 2016; Ha et al., 2017), and continual learning with biologically-realistic synapses (Kaplanis et al., 2018; Leimer et al., 2019), to name a few. Although there is no explicit separation of timescales in our weight components, the update accumulation in $\theta$ as $\phi_k$ varies (cf. Algorithm 1) suggests interpreting the base $\theta$ as slow weights and the late-phase $\Phi$ as fast weights.

This accumulation is reminiscent of a recent meta-learning algorithm (Zintgraf et al., 2019), which first separates parameters into task-shared and task-specific, and then differentiates through a sequence of accumulated updates performed over the task-specific parameters (Finn et al., 2017). Continuing with the fast-slow weight analogy, our averaging over fast weights at the end of learning (Eq. 1) could be thought of as a synaptic consolidation step which integrates the fast weight components onto a slow, persistent form of memory.

## 5    CONCLUSION

We proposed to replicate and learn in parallel a subset of weights in a late phase of neural network learning. These late-phase weights define an ensemble of models which share every other weight. We studied convolutional neural networks, a common recurrent neural network, and a simple quadratic problem. Surprisingly, across these cases, we found that a small number of appropriately chosen such weights can quickly guide SGD towards solutions that generalize well. Most of our experiments relied on BatchNorm late-phase weights, making our method easy to implement in a wide range of existing models, including pretrained ones. We expect future work to uncover new effective late-phase weight models.

## ACKNOWLEDGEMENTS

This work was supported by the Swiss National Science Foundation (B.F.G. CRSII5-173721 and 315230_189251), ETH project funding (B.F.G. ETH-20 19-01), the Human Frontiers Science Program (RGY0072/2019) and funding from the Swiss Data Science Center (B.F.G, C17-18, J.v.O. P18-03). João Sacramento was supported by an Ambizione grant (PZ00P3_186027) from the Swiss National Science Foundation. We would like to thank Nicolas Zucchet, Simon Schug, Xu He, Ângelo Cardoso and Angelika Steger for feedback, Mark van Rossum for discussions on flat minima, Simone Surace for his detailed feedback on Appendix C, and Asier Mujika for providing very useful starter code for our LSTM experiments.

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

## A  ADDITIONAL IMPLEMENTATION DETAILS

**Hypernetwork model.** The base neural network architecture we use when parameterizing our weights using a hypernetwork is identical to the WRN 28-10 described by Zagoruyko & Komodakis (2016). Our hypernetwork implementation closely follows Savarese & Maire (2019), who studied high-performing linear hypernetwork architectures for WRNs. We do not use dropout or biases in the convolutional layers. The parameters of every convolutional layer are hypernetwork-generated, with one hypernetwork per layer group (Table 7). The remaining parameters, namely those of BatchNorm units and final linear layer weights, are non-hypernetwork-generated.

Following Savarese & Maire (2019) we turn off weight decay for the model embeddings and initialize these parameters with a random pseudo-orthogonal initialization over layers. The hypernetwork parameters are initialized using a standard Kaiming initialization (He et al., 2015).

Table 7: Specification of the hypernetwork used for each convolutional layer of the WRN, indexed by its depth in the network. A depth marked by * refers to the residual connection spanning across the specified layers. The characteristics of each layer is described in the format input-channels $\times$ [kernel-size] $\times$ output-channels under *Conv-layer*. Layers within the same group are generated by the same hypernetwork. Each hypernetwork has a unique parameter tensor of shape *Hnet-PS*, which, when multiplied by a layer and weight embedding of shape *Emb-PS* and reshaped appropriately, generates the primary network parameter of shape *Base-PS*.

| Depth | Conv-layer | Base-PS | Layer group | Hnet-PS | Emb-PS |
|---|---|---|---|---|---|
| 1 | 3×[3×3]×16 | [16, 3, 3, 3] | 0 | [16, 3, 3, 3, 10] | [10, 1] |
| 2 | 16×[3×3]×160 | [160, 3, 3, 16] | 1 | [160, 3, 3, 16, 7] | [7, 1] |
| 3 | 160×[3×3]×160 | [160, 3, 3, 160] | 2 | [160, 3, 3, 80, 14] | [14, 2] |
| 4 | 160×[3×3]×160 | [160, 3, 3, 160] | 2 | [160, 3, 3, 80, 14] | [14, 2] |
| 5 | 160×[3×3]×160 | [160, 3, 3, 160] | 2 | [160, 3, 3, 80, 14] | [14, 2] |
| 6 | 160×[3×3]×160 | [160, 3, 3, 160] | 2 | [160, 3, 3, 80, 14] | [14, 2] |
| 7 | 160×[3×3]×160 | [160, 3, 3, 160] | 2 | [160, 3, 3, 80, 14] | [14, 2] |
| 8 | 160×[3×3]×160 | [160, 3, 3, 160] | 2 | [160, 3, 3, 80, 14] | [14, 2] |
| 9 | 160×[3×3]×160 | [160, 3, 3, 160] | 2 | [160, 3, 3, 80, 14] | [14, 2] |
| 10 | 160×[3×3]×320 | [320, 3, 3, 160] | 3 | [320, 3, 3, 160, 14] | [14, 1] |
| 11 | 320×[3×3]×320 | [320, 3, 3, 320] | 3 | [320, 3, 3, 160, 14] | [14, 2] |
| 12 | 320×[3×3]×320 | [320, 3, 3, 320] | 3 | [320, 3, 3, 160, 14] | [14, 2] |
| 13 | 320×[3×3]×320 | [320, 3, 3, 320] | 3 | [320, 3, 3, 160, 14] | [14, 2] |
| 14 | 320×[3×3]×320 | [320, 3, 3, 320] | 3 | [320, 3, 3, 160, 14] | [14, 2] |
| 15 | 320×[3×3]×320 | [320, 3, 3, 320] | 3 | [320, 3, 3, 160, 14] | [14, 2] |
| 16 | 320×[3×3]×320 | [320, 3, 3, 320] | 3 | [320, 3, 3, 160, 14] | [14, 2] |
| 17 | 320×[3×3]×320 | [320, 3, 3, 320] | 3 | [320, 3, 3, 160, 14] | [14, 2] |
| 18 | 320×[3×3]×640 | [640, 3, 3, 320] | 4 | [640, 3, 3, 320, 14] | [14, 1] |
| 19 | 640×[3×3]×640 | [640, 3, 3, 640] | 4 | [640, 3, 3, 320, 14] | [14, 2] |
| 20 | 640×[3×3]×640 | [640, 3, 3, 640] | 4 | [640, 3, 3, 320, 14] | [14, 2] |
| 21 | 640×[3×3]×640 | [640, 3, 3, 640] | 4 | [640, 3, 3, 320, 14] | [14, 2] |
| 22 | 640×[3×3]×640 | [640, 3, 3, 640] | 4 | [640, 3, 3, 320, 14] | [14, 2] |
| 23 | 640×[3×3]×640 | [640, 3, 3, 640] | 4 | [640, 3, 3, 320, 14] | [14, 2] |
| 24 | 640×[3×3]×640 | [640, 3, 3, 640] | 4 | [640, 3, 3, 320, 14] | [14, 2] |
| 25 | 640×[3×3]×640 | [640, 3, 3, 640] | 4 | [640, 3, 3, 320, 14] | [14, 2] |
| 2→4* | 16×[1×1]×160 | [160,1, 1, 16] | 5 | [160, 1, 1, 16, 7] | [7, 1] |
| 10→12* | 160×[1×1]×320 | [320, 1, 1, 160] | 6 | [320, 1, 1, 160, 7] | [7, 1] |
| 18→20* | 320×[1×1]×640 | [640, 1, 1, 320] | 7 | [640, 1, 1, 320, 7] | [7, 1] |

**Small ConvNet model.** We train a slight modification of the classic LeNet-5 (Lecun et al., 1998) for 200 epochs on CIFAR-10. Both convolutional and fully-connected layers are left unchanged, but we use rectified linear units on the hidden layers. Furthermore, after each such activation, Batch-Norm units are inserted. We optimize the model with SGD and use late-phase BatchNorm weights, with $T_0 = 50$ and $\sigma_0 = 0.5$. For simplicity of implementation, we do not include the last linear layer in the late-phase weight set $\Phi$.

**Optimization.** We optimize the cross-entropy loss, using either SGD with Nesterov momentum (0.9) or SGD with Nesterov momentum (0.9) wrapped inside SWA. **LSTM:** Our LSTM experiments use Adam with constant learning rate 0.001, batch size 128, and no regularizers such as weight decay or dropout. **WRN-28-10:** For our WRN experiments on the CIFAR datasets we use the learning rate annealing schedule of Izmailov et al. (2018), according to which an initial learning rate of 0.1 is linearly decreased at every epoch from the end of the 100th epoch (80th for SWA) to the end of the 180th epoch (144th for SWA; SWA is activated at epoch 160), when a final value of 0.001 (0.05 for SWA) is reached. Our optimizers use Nesterov momentum (set to 0.9), a batch size of 128 and weight decay (set to 0.0005). On CIFAR-100 (SGD) we set the weight decay of late-phase weights proportional to the ensemble size, $0.0005K$. **WRN-28-14:** The WRN 28-14 models are trained for 300 epochs on CIFAR-100. The learning rate is initialized at 0.1, then annealed to 0.05 from the 80th epoch to the 240th epoch. SWA is activated at epoch 160. All other hyperparameters are identical to those of WRN 28-10. **ConvNet:** Same as for the WRN 28-10 model, except that we anneal the learning rate until the 160th epoch.

---

**Algorithm 2:** Stochastic weight averaging (SWA)

---

**Require:** Base weights $\theta$, dataset $\mathcal{D}$,
  hyperparameter $\eta$, loss $\mathcal{L}$
**Require:** Training iteration $t$
$\mathcal{M} \leftarrow$ Sample minibatch from $\mathcal{D}$
$\Delta\theta \leftarrow \nabla_\theta \mathcal{L}(\mathcal{M}, \theta)$
$\theta \leftarrow U(\theta, \eta, \Delta\theta)$
$\theta_{\text{SWA}} \leftarrow (t\,\theta_{\text{SWA}} + \theta)/(t+1)$
$t \leftarrow t + 1$

---

**Algorithm 3:** SGD with Nesterov momentum

---

**Require:** Base weights $\theta$, dataset $\mathcal{D}$,
  learning rate $\eta$, momentum $\rho$,
  loss $\mathcal{L}$
$\mathcal{M} \leftarrow$ Sample minibatch from $\mathcal{D}$
$\Delta\theta \leftarrow \nabla_\theta \mathcal{L}(\mathcal{M}, \theta + \rho\,\nu)$
$\nu \leftarrow \rho\nu - \eta\,\Delta\theta$
$\theta \leftarrow \theta + \nu$

---

Figure 4: Pseudocode for a single parameter update for SWA and SGD with Nesterov momentum, the two main optimizers used in our experiments. These are either used standalone, or as $U_\theta$ and $U_\phi$ in Algorithm 1 (main text). $U$ in Algorithm 2 (SWA) serves as a placeholder for a parameter update rule such as SGD (with Nesterov momentum) or Adam. Training iteration $t$ is counted from the activation of SWA in Algorithm 1.

**Batch normalization units.** Whenever we use SWA, we follow Izmailov et al. (2018) and perform a full pass over the training set to re-estimate BatchNorm unit statistics before testing. This correction is required since the online BatchNorm mean and variance estimates track the activations produced with the raw (non-averaged) weights during training, while the averaged solution is the one used when predicting at test time.

**Data augmentation and preprocessing.** On both CIFAR and ImageNet datasets, all images are normalized channelwise by subtracting the mean and dividing by the standard deviation; both statistics are computed on the training dataset. The same transformation is then applied when testing, including to OOD data. Following a standard procedure (e.g., Zagoruyko & Komodakis, 2016; He et al., 2016) we augment our training datasets using random crops (with a 4-pixel padding for CIFAR) and random horizontal flips. The ImageNet training dataset is augmented with random horizontal flips, as well as random cropping of size 224, while a centered cropping of size 224 was used on the test set. Our OOD datasets are resized to fit whenever necessary; we used the resized images made available by Lee et al. (2018).

**ImageNet experiments.** The pretrained model for the ImageNet experiment is obtained from torchvision's models subpackage. We fine-tune the model for 20 additional epochs on ImageNet.

---

**Algorithm 4:** Late-phase learning

---

**Require:** Base weights $\theta$, late-phase weight set $\Phi$, dataset $\mathcal{D}$, gradient scale
        factor $\gamma_\theta$, learning rate $\eta$, ensemble size $K$, initialization noise
        $\sigma_0$, initialization time $T_0$, number of training iterations $T$, loss $\mathcal{L}$

**Initialization:** $\hat{K} \leftarrow 0$, $s \leftarrow 0$, $t \leftarrow 1$

**while** $t \leq T$ **do**
    **if** $t = T_0$ **then**
        // generate late-phase weights
        **for** $1 \leq k \leq K$ **do**
            sample $\epsilon \sim \mathcal{N}(0,1)$
            $\phi_k \leftarrow \phi_0 + \frac{\sigma_0}{\|\phi_0\|}\epsilon$
        // set range for specialists training
        $\hat{K} \leftarrow K$
        $s \leftarrow 1$
    **for** $s \leq k \leq \hat{K}$ **do**
        $\mathcal{M}_k \leftarrow$ Sample minibatch from $\mathcal{D}$
        $\Delta\theta_k \leftarrow \nabla_\theta \mathcal{L}(\mathcal{M}_k, \theta, \phi_k)$
        $\phi_k \leftarrow \phi_k - \eta \nabla_{\phi_k} \mathcal{L}(\mathcal{M}_k, \theta, \phi_k)$
        $t \leftarrow t + 1$
    $\theta \leftarrow \theta - \gamma_\theta \, \eta \sum_{k=1}^{K} \Delta\theta_k$

---

Figure 5: Complete pseudocode for an entire training session using late-phase weights. To avoid notational clutter $T$, $T_0$ and $t$ are measured in numbers of minibatches consumed. In the paper, we measure $T_0$ and $T$ in epochs. For simplicity, we present the case where $U_\phi$ and $U_\theta$ are set to plain SGD (without momentum) and $\phi_k$ of dimension 1. Other optimization algorithms (e.g., Algorithm 2 or Algorithm 3) can be used to replace $U_\phi$ and $U_\theta$, as described in Algorithm 1. Note that we increase $t$ inside the inner loop. This highlights (*i*) that every specialist parameter is trained only on $1/K$ data samples after $t > T_0$ compared to $\theta$, and (*ii*) that we count every minibatch drawn from the data to compare fairly to algorithms without an inner loop.

We use a multistep learning rate scheduler, starting at 0.001 then decreasing at the 10th epoch to 0.0001. We use SGD with momentum (set to 0.9) and weight decay (set to 0.0001) as our optimizer, with a batch size of 256. We use $\sigma_0 = 0$ and $K = 10$ for our late-phase model.

**Code forks.** Our hypernetwork implementation was inspired by the code made publicly available by Savarese & Maire (2019). Our implementation of SWA was adapted from the code accompanying the work of Izmailov et al. (2018), now available on the torchcontrib Python package. The SWAG method was evaluated directly using the code provided by the authors (Maddox et al., 2019). We used the same base WRN model as Maddox et al. (2019), which can be retrieved from `https://github.com/meliketoy/wide-resnet.pytorch`.

Table 8: CIFAR-100 test set accuracy (%) depending on different values of $K$ for WRN 28-10, SGD. Mean $\pm$ std. over 5 seeds.

| $K$ | Test acc. (%) |
|---|---|
| 1 | $81.35^{\pm 0.16}$ |
| 5 | $82.44^{\pm 0.22}$ |
| 10 | $82.87^{\pm 0.22}$ |
| 15 | $83.01^{\pm 0.27}$ |
| 20 | $82.86^{\pm 0.29}$ |

**LSTM** All experiments are conducted using the Tensorflow Python framework (Abadi et al., 2016). All base weights are initialized uniform in $[-0.01, 0.01]$ whereas the initial rank-1 matrix weights are centered around 1 i.e. $[1-0.01, 1+0.01]$ to mitigate strong difference in initialization compared to the base model. We use the Tensorflow default values ($\beta_1 = 0.9$, $\beta_2 = 0.$, $\epsilon = 10^{-8}$) for the Adam optimiser. We perform a grid search over $\sigma_0 \in [0, 0.5]$ (in steps of size 0.05) for our LSTM experiments (fixing $K = 10$ and varying $T_0 \in \{0, 30\}$) and obtain the values reported in the main text, $T_0 = 30$ and $\sigma_0 = 0.35$.

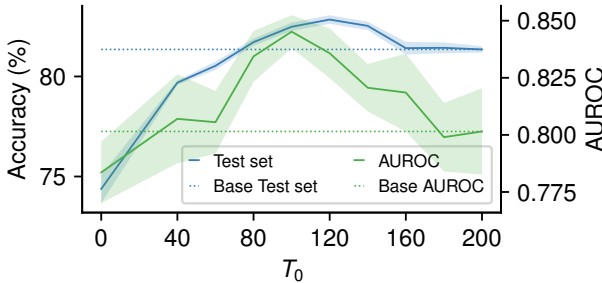

Figure 6: Sensitivity analysis of $T_0$. Mean AUROC score (OOD) and test set accuracy for different values of $T_0$ for WRN 28-10, CIFAR-100, SGD, with BatchNorm late-phase weights.

## B  ADDITIONAL EXPERIMENTS

Table 9: Applying late-phase weights to a pretrained WRN 28-10, CIFAR-100, SGD. Mean $\pm$ std. over 5 seeds.

| Model | Test acc. (%) |
|---|---|
| Initial | $81.35^{\pm 0.16}$ |
| Base | $81.47^{\pm 0.14}$ |
| Late-phase BatchNorm | $82.02^{\pm 0.12}$ |
| Late-phase BatchNorm, frozen base weight | $81.50^{\pm 0.20}$ |

**Pretrained CIFAR-100.**  We apply our method to a standard WRN 28-10 pretrained on CIFAR-100 (i.e., we set $T_0 = 200$) and train for an additional 20 epochs. At the beginning of the fine-tuning, the learning rate is reset to 0.01, then annealed linearly to 0.001 for 10 epochs. It is then held constant for the remainder of the fine-tuning process. We observe that augmenting with BatchNorm late-phase weights yields an improved predictive accuracy compared to additional fine-tuning with SGD (Base), cf. Table 9. Both methods improve over the initial baseline (Initial), including the base model. This can be explained by the optimization restart and the accompanying spike in the learning rate introduced by our scheduler (Loshchilov & Hutter, 2017).

Importantly, we find that fine-tuning only BatchNorm late-phase weights while keeping all other weights fixed does not even match the Base control. Together with the finding that the optimal late-phase weight initialization time is at $T_0^* = 120$ (when learning for 200 epochs), this result speaks to the importance of jointly optimizing both base and late-phase weights through our Algorithm 1.

**Gradient accumulation control.**  Here we show that the improved generalization we report in the main text is not merely due to gradient accumulation over larger batches. We take our base WRN 28-10 model (without late-phase weights) and start accumulating gradients over $K = 10$ minibatches at $T_0 = 120$, experimenting both with $\gamma_\theta = 1/K$ and $\gamma_\theta = 1$. The models are trained with SGD using otherwise standard optimization settings. Both controls fail to improve (even match) the performance of the base model trained without any gradient accumulation.

**Sensitivity to $T_0$, $K$ and $\sigma_0$.**  We present a hyperparameter exploration on the CIFAR-100 dataset using BatchNorm late-phase weights in Tables 8, 11 and 12. We find that our algorithm is largely

Table 10: Gradient accumulation control, CIFAR-100, WRN 28-10, SGD. Mean $\pm$ std. over 5 seeds.

| Model | Test acc. (%) |
|---|---|
| Base (SGD) | $81.35^{\pm 0.16}$ |
| Base + gradient accumulation ($\gamma_\theta = 1$) | $80.76^{\pm 0.26}$ |
| Base + gradient accumulation ($\gamma_\theta = 1/K$) | $80.34^{\pm 0.28}$ |

Table 11: CIFAR-100 test set accuracy (%) depending on different values of $\sigma_0$ for WRN 28-10 SGD with late-phase BatchNorm weights (LPBN). Mean $\pm$ std. over 5 seeds.

| $\sigma_0$ | CIFAR-100 (LPBN) | | CIFAR-100 (LPBN, non-averaged) | | CIFAR-100 (LPBN, pretrained) | |
|---|---|---|---|---|---|---|
| | Test acc. (%) | OOD | Test acc. (%) | OOD | Test acc. (%) | OOD |
| 0 | $82.87^{\pm0.22}$ | $0.833^{\pm0.005}$ | $83.20^{\pm0.20}$ | $0.854^{\pm0.017}$ | $81.70^{\pm0.19}$ | $0.803^{\pm0.017}$ |
| 0.25 | $82.77^{\pm0.19}$ | $0.836^{\pm0.012}$ | $82.68^{\pm0.32}$ | $0.861^{\pm0.013}$ | $82.02^{\pm0.12}$ | $0.808^{\pm0.017}$ |
| 0.5 | $82.78^{\pm0.18}$ | $0.837^{\pm0.011}$ | $82.71^{\pm0.10}$ | $0.862^{\pm0.009}$ | $81.15^{\pm0.29}$ | $0.797^{\pm0.007}$ |
| 0.75 | $82.41^{\pm0.20}$ | $0.839^{\pm0.012}$ | $82.43^{\pm0.15}$ | $0.855^{\pm0.013}$ | - | - |
| 1.0 | $81.52^{\pm1.09}$ | $0.840^{\pm0.017}$ | $82.38^{\pm0.15}$ | $0.848^{\pm0.014}$ | - | - |

robust to $\sigma_0$ when $T_0$ can be set to its optimal value, which is at 60% of training. See also Figure 6 for a visualisation of the same data, specifically the change in mean AUROC score and test set accuracy when changing $T_0$. This result holds also on CIFAR-10, cf. Table 12. When starting from a pretrained condition ($T_0 = 200$), finite $\sigma_0$ leads to a significant improvement in performance, cf. Table 11. We therefore report results obtained with $\sigma_0 = 0$ for every CIFAR and ImageNet experiment in the main text. The exception to this is the non-averaged (ensemble) late-phase BatchNorm model presented in Table 4, which was optimized for best OOD performance (corresponding to $\sigma_0 = 0.5$).

Table 13: Performance of a WRN 28-10 on CIFAR-100 with different dropout probability $p$. For MC-dropout we average over 10 different samples. Mean $\pm$ std. over 5 seeds.

| | $p$ | Test acc. (%) | Test NLL | OOD |
|---|---|---|---|---|
| Dropout | 0.1 | $81.46^{\pm0.13}$ | $0.7476^{\pm0.0059}$ | $0.8031^{\pm0.0064}$ |
| Dropout | 0.2 | $81.31^{\pm0.20}$ | $0.7736^{\pm0.0025}$ | $0.8022^{\pm0.0299}$ |
| Dropout | 0.3 | $80.93^{\pm0.19}$ | $0.8342^{\pm0.0098}$ | $0.7833^{\pm0.0239}$ |
| MC-Dropout | 0.1 | $81.51^{\pm0.14}$ | $0.7197^{\pm0.0054}$ | $0.8149^{\pm0.0087}$ |
| MC-Dropout | 0.2 | $81.55^{\pm0.11}$ | $0.7105^{\pm0.0026}$ | $0.8225^{\pm0.0488}$ |
| MC-Dropout | 0.3 | $81.36^{\pm0.31}$ | $0.7150^{\pm0.0069}$ | $0.8040^{\pm0.0135}$ |

**Related work.** Here we provide details for the training setups of alternative methods we compare against in the main text. For the results reported for dropout (Srivastava et al., 2014) and MC-dropout (Gal & Ghahramani, 2016), we simply train a WRN 28-10 on CIFAR-100 with the exact same configuration as for our base model, see above, but include dropout layers as usually done (Zagoruyko & Komodakis, 2016) after the first convolution in each residual block. For a scan over the dropout probability $p$ in this setup, see Table 13. $p = 0.2$ is reported in the main text - for CIFAR-100 and CIFAR-10. Note that $p$ was only tuned for CIFAR-100.

For the reported results of BatchEnsemble (Wen et al., 2020), we simply execute the code provided by the authors at `https://github.com/google/uncertainty-baselines` with their fine-tuned configuration for CIFAR-10/100. No-

Table 12: CIFAR-10 and CIFAR-100 test set accuracy (%) depending on different late phase timing $T_0$ for WRN 28-10, SGD. Mean $\pm$ std. over 5 seeds.

| $T_0$ | CIFAR-10 | CIFAR-100 |
|---|---|---|
| 0 | $95.68^{\pm0.23}$ | $74.38^{\pm0.71}$ |
| 40 | $96.34^{\pm0.08}$ | $79.69^{\pm0.11}$ |
| 60 | $96.42^{\pm0.10}$ | $80.53^{\pm0.21}$ |
| 80 | $96.50^{\pm0.11}$ | $81.72^{\pm0.18}$ |
| 100 | $96.45^{\pm0.08}$ | $82.48^{\pm0.21}$ |
| 120 | $96.48^{\pm0.20}$ | $82.87^{\pm0.22}$ |
| 140 | $96.26^{\pm0.17}$ | $82.53^{\pm0.21}$ |
| 160 | $96.23^{\pm0.11}$ | $81.41^{\pm0.31}$ |
| 180 | $96.25^{\pm0.23}$ | $81.43^{\pm0.27}$ |
| 200 | $96.16^{\pm0.12}$ | $81.35^{\pm0.16}$ |

tably, the authors use a different setup than followed in this manuscript. First, the WRN 28-10 is trained for 250 epochs (we allow for this increased budget exceptionally for BatchEnsemble), with a multi-step learning rate annealing at $[80, 160, 180]$ with a learning rate decay factor of 0.2. Second, a weight decay of $3 \times 10^{-4}$ is used.

Table 14: Final training set loss on CIFAR datasets, WRN 28-10, SGD. Mean $\pm$ std. over 5 seeds.

|  | Training loss |
|---|---|
| CIFAR-10 – base | $0.0010^{\pm 0.0000}$ |
| CIFAR-10 – late-phase BatchNorm | $0.0019^{\pm 0.0001}$ |
| CIFAR-100 – base | $0.0024^{\pm 0.0001}$ |
| CIFAR-100 – late-phase BatchNorm | $0.0267^{\pm 0.0004}$ |

Table 15: Performance of models trained on a reduced CIFAR-10 training set and evaluated on the full CIFAR-10 test set. Mean $\pm$ std. over 5 seeds.

| Model | Test acc. (%) |
|---|---|
| Base (SGD) | $88.98^{\pm 0.18}$ |
| Late-phase BN (SGD) | $89.58^{\pm 0.19}$ |

For the results reported for SWAG (Maddox et al., 2019), we use the code provided by the authors at `https://github.com/wjmaddox/swa_gaussian`, and the proposed fine-tuned configuration which coincides with the configuration used to obtain all CIFAR-100 results reported in this manuscript, except for BatchEnsembles (see above). We report results for SWAG after training on 200 epochs for fair comparison.

**Training losses.** We provide the final achieved training losses for the base model and when augmenting it with BatchNorm late-phase weights on Table 14, for both CIFAR-10 and CIFAR-100. Using a fast gradient accumulation scale factor of $\gamma_\theta = 1$ leads to a higher training loss on CIFAR-100 than that of the standard model, but we found this setting crucial to achieve the largest improvement on test set generalization.

**CIFAR-10 with a reduced training set.** Here we evaluate the performance of our method on a reduced training set of CIFAR-10. We randomly pick 10000 training data out of the 50000 available, and use this new set to train different models. After training, the models are evaluated on the standard CIFAR-10 test set. Results are shown in Table 15.

**Detailed OOD results and mean corruption error (mCE) experiments.** In order to test the robustness of late-phase weights against input data corruption, we used the corruptions and dataset proposed by Hendrycks & Dietterich (2019), freely available at `https://github.com/hendrycks/robustness`. The authors propose 15 noise sources such as random Gaussian noise, spatter or contrast changes to deform the input data and report the model test set accuracy on the corrupted dataset under 5 severity levels (noise strengths). For each source noise, its corruption error is computed by averaging the prediction error over the severity levels. The average of the corruption error of all 15 noises gives us the Mean Corruption Error (mCE). See Table 16 for the mCE computed on the corrupted CIFAR-100 dataset.

**Training run time.** Here we compare the training run time of our method with the baseline. The result was computed in Python 3.7, using the automatic differentiation and GPU acceleration package PyTorch (version 1.4.0). We used the standard datasets (including training and test splits) as provided by the torchvision package unless stated otherwise. We used a single NVIDIA GeForce 2080 Ti GPU for the experiment. Results are presented in Table 17.

## C   THEORETICAL ANALYSIS OF THE NOISY QUADRATIC PROBLEM

In this section, we consider a noisy quadratic problem (NQP) that can be theoretically analyzed and that captures important characteristics of the stochasticity of a minibatch-based optimizer (Schaul et al., 2013; Martens, 2016; Wu et al., 2018; Zhang et al., 2019a;b). The NQP does a second-order Taylor expansion of the loss function around the optimum $\mathbf{w}^*$ and models the minibatch noise as

Table 16: OOD performance measured by the AUROC, and robustness measured by the Mean Corruption Error (*mCE*). We train the models on CIFAR-100 and attempt to discriminate test set images from novel ones drawn from the SVHN, LSUN, Tiny ImageNet (*TIN*) and CIFAR-10 dataset. The mCE value is the average across 75 different corruptions from the CIFAR-100-C dataset. LPBN and LP HNET stand respectively for late-phase BatchNorm and late-phase hypernetwork.

| | SVHN | LSUN | TIN | CIFAR-10 | mCE |
|---|---|---|---|---|---|
| Base | $0.814^{\pm 0.024}$ | $0.798^{\pm 0.036}$ | $0.776^{\pm 0.038}$ | $0.818^{\pm 0.003}$ | $47.84^{\pm 0.41}$ |
| LPBN | $0.831^{\pm 0.021}$ | $0.862^{\pm 0.017}$ | $0.838^{\pm 0.023}$ | $0.814^{\pm 0.002}$ | $45.59^{\pm 0.25}$ |
| LPBN (non-avg.) | $0.877^{\pm 0.008}$ | $0.883^{\pm 0.015}$ | $0.863^{\pm 0.023}$ | $0.827^{\pm 0.002}$ | $46.21^{\pm 0.29}$ |
| LP HNET | $0.815^{\pm 0.022}$ | $0.842^{\pm 0.023}$ | $0.816^{\pm 0.027}$ | $0.811^{\pm 0.002}$ | $47.84^{\pm 0.42}$ |
| Dropout (Mean) | $0.792^{\pm 0.093}$ | $0.807^{\pm 0.040}$ | $0.788^{\pm 0.044}$ | $0.822^{\pm 0.003}$ | $48.97^{\pm 0.33}$ |
| MC-Dropout | $0.806^{\pm 0.082}$ | $0.842^{\pm 0.046}$ | $0.817^{\pm 0.041}$ | $0.824^{\pm 0.003}$ | $48.09^{\pm 0.36}$ |
| SWAG | $0.824^{\pm 0.012}$ | $0.839^{\pm 0.054}$ | $0.835^{\pm 0.041}$ | $0.816^{\pm 0.004}$ | - |
| BatchEnsemble | $0.848^{\pm 0.020}$ | $0.828^{\pm 0.018}$ | $0.820^{\pm 0.030}$ | $0.829^{\pm 0.019}$ | - |
| Deep ens. | 0.839 | 0.836 | 0.812 | 0.839 | 44.21 |
| Deep ens. (LPBN) | 0.855 | 0.884 | 0.856 | 0.834 | 43.15 |

Table 17: Training time in seconds and hours on CIFAR-10 for 200 epochs on a single NVIDIA GeForce 2080 Ti GPU.

| Model | seconds | hours |
|---|---|---|
| Base (SGD) | 17714 | $\sim 4.92$ |
| Late-phase BN (SGD) | 17772 | $\sim 4.94$ |

a random translation $\epsilon$ of the optimum, while keeping the curvature $H$ the same. This gives us the following minibatch loss:

$$\hat{\mathcal{L}} = \frac{1}{2}(\mathbf{w} - \mathbf{w}^* + \frac{1}{\sqrt{B}}\epsilon)^T H(\mathbf{w} - \mathbf{w}^* + \frac{1}{\sqrt{B}}\epsilon) \qquad (4)$$

with $\epsilon \sim \mathcal{N}(\mathbf{0}, \Sigma)$ and $B$ the minibatch size. Note that we use boldface notation for vectors in this analysis for notational clarity. The NQP can be seen as an approximation of the loss function in the final phase of learning, where we initialize the late-phase ensemble. Despite its apparent simplicity, it remains a challenging optimization problem that has important similarities with stochastic mini-batch training in deep neural networks (Schaul et al., 2013; Martens, 2016; Wu et al., 2018; Zhang et al., 2019a;b). For the simple loss landscape of the NQP, there are three main strategies to improve the expected loss after convergence: (i) increase the mini-batch size $B$ (Zhang et al., 2019a), (ii) use more members $K$ in an ensemble (c.f. Section C.3 and (iii) decrease the learning rate $\eta$ (Schaul et al., 2013; Martens, 2016; Wu et al., 2018; Zhang et al., 2019a;b). The late-phase weights training combines the two first strategies in a non-trivial manner by (i) averaging over the base-weights gradients for all ensemble members and (ii) averaging the late-phase weights in parameter space to obtain a mean-model. The goal of this theoretical analysis is to show that the expected loss after convergence scales inversely with the number of late-phase ensemble members $K$, which indicates that the non-trivial combination of the two strategies is successful.

To model the multiplicative weight interaction between late-phase weights and base weights, we use linear hypernetworks of arbitrary dimension. The linear hypernetworks parameterize the weights as $\mathbf{w} = \theta\mathbf{e}$, with $\theta \in \mathbb{R}^{n \times d}$ the hypernetwork parameters and $\mathbf{e} \in \mathbb{R}^d$ the embedding vector. The embedding vectors $\mathbf{e}$ are used as late-phase weights ($\phi$ in the main manuscript) to create a late-phase ensemble with $K$ members, while using a shared hypernetwork $\theta$ as base-weights: $\mathbf{w}_k = \theta\mathbf{e}_k$. Ultimately, we are interested in the expected risk of the the mean model at steady state:

$$\mathbb{E}[\mathcal{L}^{(ss)}] = \mathbb{E}_{\rho_{ss}}[\frac{1}{2}(\bar{\mathbf{w}} - \mathbf{w}^*)^T H(\bar{\mathbf{w}} - \mathbf{w}^*)] \qquad (5)$$

with $\bar{\mathbf{w}} \triangleq \frac{1}{K}\sum_k \theta\mathbf{e}_k = \theta\frac{1}{K}\sum_k \mathbf{e}_k \triangleq \theta\bar{\mathbf{e}}$ and $\rho_{ss}$ the steady-state distribution of the parameters. Note that we cannot put $\mathbf{w}^* = \mathbf{0}$ without loss of generality, because the overparameterization of the hypernetworks makes the optimization problem nonlinear.

We start with investigating the discrete time dynamics induced by late-phase learning, after which we derive the corresponding continuous time dynamics to be able to use the rich stochastic dynamical systems literature for analyzing the resulting nonlinear stochastic dynamical system.

## C.1 DISCRETE TIME DYNAMICS

As we want to investigate the multiplicative interaction between the shared and late-phase parameters, we substitute $\mathbf{w} = \theta\mathbf{e}$ into equation 4, instead of computing a new Taylor approximation in the hypernetwork parameter space. Let us take $t$ as the index for the outer loop (updating $\theta$) and $k$ the index for the ensemble member. Then we have the following stochastic minibatch loss:

$$\hat{\mathcal{L}}^{(t,k)} = \frac{1}{2}(\theta^{(t)}\mathbf{e}_k^{(t)} - \mathbf{w}^* + \frac{1}{\sqrt{B}}\boldsymbol{\epsilon}^{(t,k)})^T H(\theta^{(t)}\mathbf{e}_k^{(t)} - \mathbf{w}^* + \frac{1}{\sqrt{B}}\boldsymbol{\epsilon}^{(t,k)}), \tag{6}$$

which gives rise to the following parameter updates using late-phase learning with learning rate $\eta$ and minibatch size $B$:

$$\theta^{(t+1)} = \theta^{(t)} - \eta\frac{1}{K}\sum_k H(\theta^{(t)}\mathbf{e}_k^{(t)} - \mathbf{w}^*)\mathbf{e}_k^{(t)T} + \frac{\eta}{\sqrt{B}}\frac{1}{K}\sum_k H\boldsymbol{\epsilon}^{(t,k)}\mathbf{e}_k^{(t)T} \tag{7}$$

$$\mathbf{e}_k^{(t+1)} = \mathbf{e}_k^{(t)} - \eta\theta^{(t)T}H(\theta^{(t)}\mathbf{e}_k^{(t)} - \mathbf{w}^*) + \frac{\eta}{\sqrt{B}}\theta^{(t)T}H\boldsymbol{\epsilon}^{(t,k)} \tag{8}$$

The above discrete time dynamics are nonlinear, giving rise to a non-Gaussian parameter distribution $\rho$. Hence, it is not possible to characterize these dynamics by the moment-propagating equations of the first and second moment as done in Zhang et al. (2019a;b); Schaul et al. (2013) and Wu et al. (2018), without having full access of the parameter distribution $\rho$. Furthermore, because of the hypernetwork parameterization, we cannot decouple the system of equations, even if $H$ and $\Sigma$ are diagonal, which is a common approach in the literature. Therefore, we investigate the corresponding continuous time dynamics, such that we can use the rich literature on stochastic dynamical systems.

## C.2 CONTINUOUS TIME DYNAMICS

First, let us define some compact notations for the various parameters.

$$\mathbf{e}_t \triangleq [\mathbf{e}_1^{(t)T} \ldots \mathbf{e}_K^{(t)T}]^T \tag{9}$$

$$E_t \triangleq [\mathbf{e}_1^{(t)} \ldots \mathbf{e}_K^{(t)}] \tag{10}$$

$$\boldsymbol{\theta}_t \triangleq \text{vec}(\theta_t) \tag{11}$$

$$\mathbf{x}_t \triangleq [\boldsymbol{\theta}_t^T, \mathbf{e}_t^T]^T \tag{12}$$

$$\boldsymbol{\epsilon}_t \triangleq [\boldsymbol{\epsilon}^{(t,1)T} \ldots \boldsymbol{\epsilon}^{(t,K)T}]^T, \tag{13}$$

$$\tag{14}$$

where $\text{vec}(\theta)$ concatenates the columns of $\theta$ in a vector. Then the discrete time dynamics (equation 7 and equation 8) can be rewritten as:

$$\mathbf{x}_{t+1} = \mathbf{x}_t - \eta F(\mathbf{x}_t) + \frac{\eta}{\sqrt{B}}G(\mathbf{x}_t)\boldsymbol{\epsilon}_t \tag{15}$$

with

$$F(\mathbf{x}_t) \triangleq \begin{bmatrix} \frac{1}{K}\sum_k (\mathbf{e}_k^{(t)} \otimes H)(\theta_t\mathbf{e}_k^{(t)} - \mathbf{w}^*) \\ (I \otimes (\theta_t^T H\theta_t))\mathbf{e}_t - \mathbb{1} \otimes (\theta_t^T H\mathbf{w}^*) \end{bmatrix} \tag{16}$$

$$G(\mathbf{x}_t \triangleq \begin{bmatrix} \frac{1}{K}E_t \otimes H \\ I \otimes (\theta_t^T H) \end{bmatrix} \tag{17}$$

$$\tag{18}$$

with $\otimes$ the Kronecker product, $I$ an identity matrix of the appropriate size and $\mathbb{1}$ a vector full of ones of the appropriate size. As a linear transformation of Gaussian variables remains a Gaussian variable, we can rewrite eq. equation 15 as follows:

$$\mathbf{x}_{t+1} = \mathbf{x}_t - \eta F(\mathbf{x}_t) + \frac{\eta}{\sqrt{B}}D(\mathbf{x}_t)\boldsymbol{\zeta}_t \tag{19}$$

with $D(x_t) \triangleq \big(G(x_t)(I \otimes \Sigma)G(x_t)^T\big)^{0.5}$ and $\zeta \sim \mathcal{N}(0, I)$. Following Liu & Theodorou (2019) and Chaudhari & Soatto (2018), the corresponding continuous-time dynamics are:

$$\mathbf{dx}_t = -F(\mathbf{x}_t)\mathrm{d}t + \sqrt{2\beta^{-1}}D(\mathbf{x}_t)\mathrm{d}\mathbf{W}_t \tag{20}$$

with $\mathbf{W}_t$ Brownian motion and $\beta \triangleq \frac{2B}{\eta}$ the inverse temperature. Note that $\sqrt{\eta}$ is incorporated in the noise covariance, such that the correct limit to stochastic continuous time dynamics can be made (Liu & Theodorou, 2019; Chaudhari & Soatto, 2018; but see Yaida, 2018). For computing the expected loss $\mathbb{E}[\mathcal{L}_t]$ of the mean model, we need to have the stochastic dynamics of this loss. Using the Itô lemma (Itô, 1951; Liu & Theodorou, 2019), which is an extension of the chain rule in the ordinary calculus to the stochastic setting, we get

$$\mathrm{d}\mathcal{L}(\mathbf{x}_t) = \Big[ -\nabla\mathcal{L}(\mathbf{x}_t)^T F(\mathbf{x}_t) + \frac{1}{2}\mathrm{Tr}\big[\tilde{D}H_\mathcal{L}\tilde{D}\big]\Big]\mathrm{d}t + \big[\nabla\mathcal{L}(\mathbf{x}_t)^T\tilde{D}\big]\mathrm{d}\mathbf{W}_t \tag{21}$$

with $\tilde{D} \triangleq \sqrt{2\beta^{-1}}D(\mathbf{x}_t)$ for notational simplicity and $H_\mathcal{L}$ the Hessian of $\mathcal{L}$ w.r.t. $\mathbf{x}_t$. As we are interested in the expected risk (equation 5), we can take the expectation of equation 21 over the parameter distribution $\rho_t(\mathbf{x})$ to get the dynamics of the first moment of the loss (also known as the backward Kolmogorov equation (Kolmogorov, 1931)):

$$\mathrm{d}\mathbb{E}_{\rho_t}\big[\mathcal{L}(\mathbf{x}_t)\big] = \mathbb{E}_{\rho_t}\Big[ -\nabla\mathcal{L}(\mathbf{x}_t)^T F(\mathbf{x}_t) + \frac{1}{2}\mathrm{Tr}\big[\tilde{D}^2 H_\mathcal{L}\big]\Big]\mathrm{d}t \tag{22}$$

In order to obtain the dynamics of the parameter distribution, the Fokker-Planck equation can be used (Jordan et al., 1998). However, due to the nonlinear nature of the stochastic dynamical system, the distribution is non-Gaussian and it is not possible (to our best knowledge) to obtain an analytical solution for equation 22. Nevertheless, we can still gain important insights by investigating the steady-state of equation 22. After convergence, the left-hand side (LHS) is expected to be zero. Hence, we have that

$$\mathbb{E}_{\rho^{ss}}\big[\nabla\mathcal{L}(\mathbf{x}_{ss})^T F(\mathbf{x}_{ss})\big] = \frac{1}{2}\mathbb{E}_{\rho^{ss}}\big[\mathrm{Tr}[\tilde{D}^2 H_\mathcal{L}]\big] \tag{23}$$

The remainder of our arguments is structured as follows. First, we will show that the left-hand-side (LHS) of equation 23 is the expectation of an approximation of a weighted norm of the gradient $\nabla\mathcal{L}$, after which we will connect this norm to the loss $\mathcal{L}$ of the mean model. Second, we will investigate the RHS to show that the late-phase learning with ensembles lowers the expected risk of the NQP at steady-state. For clarity and ease of notation, we will drop the $ss$ subscripts. The gradient of the mean-model loss is given by:

$$\nabla\mathcal{L}(\mathbf{x}) = \begin{bmatrix} \big(\bar{\mathbf{e}} \otimes H\big)\big(\theta\bar{\mathbf{e}} - \mathbf{w}^*\big) \\ \frac{1}{K}\mathbb{1} \otimes \big(\theta^T H\theta\bar{\mathbf{e}} - \mathbf{w}^*\big) \end{bmatrix} \tag{24}$$

By introducing $\Delta\mathbf{e}_k \triangleq \mathbf{e}_k - \bar{\mathbf{e}}$ and using that $\sum_k \Delta\mathbf{e}_k = 0$, we can rewrite $F(\mathbf{x})$ as:

$$F(\mathbf{x}) = \begin{bmatrix} I & 0 \\ 0 & KI \end{bmatrix} \nabla\mathcal{L}(\mathbf{x}) + \begin{bmatrix} (\Gamma \otimes H)\boldsymbol{\theta} \\ \big(I \otimes (\theta^T H\theta)\big)\Delta\mathbf{e} \end{bmatrix} \tag{25}$$

with $\Gamma \triangleq \frac{1}{K}\sum_k \Delta\mathbf{e}_k\Delta\mathbf{e}_k^T$ and $\Delta\mathbf{e}^T \triangleq [\mathbf{e}_1^T...\mathbf{e}_K^T]$. We see that $F$ is an approximation of the gradient $\nabla\mathcal{L}$ where the lower block of $\nabla\mathcal{L}$ is scaled by $K$. Importantly, the lower block of the second element of the RHS of equation 25 (the approximation error) will disappear when taking the inner product with $\nabla\mathcal{L}$ and the upper block is not influenced by the number of ensemble members $K$, which we will need later. The LHS of equation 23 can now be rewritten as:

$$\mathbb{E}_{\rho^{ss}}\big[\nabla\mathcal{L}(\mathbf{x})^T F(\mathbf{x})\big] = \mathbb{E}_{\rho^{ss}}\big[\nabla\mathcal{L}(\mathbf{x})^T M\nabla\mathcal{L}(\mathbf{x})\big] + \mathbb{E}_{\rho^{ss}}\big[\mathrm{Tr}[H\theta\Gamma H(\theta\bar{\mathbf{e}} - \mathbf{w}^*)\bar{\mathbf{e}}^T]\big] \tag{26}$$

with $M$ the diagonal matrix of equation 25 (first element of the RHS). The first term of the RHS of equation 26 is the expectation of a weighted squared norm of $\nabla\mathcal{L}$, while the second term is an approximation error due to the covariance of $\Delta\mathbf{e}_k$. Hence, we see that the LHS of equation 23 can be seen as an approximation of a weighted norm of the gradient $\nabla\mathcal{L}$. By investigating the term $\nabla\mathcal{L}(\mathbf{x})^T M\nabla\mathcal{L}(\mathbf{x})$ further, we show that it is closely connected to the loss $\mathcal{L}$.

$$\nabla\mathcal{L}(\mathbf{x})^T M\nabla\mathcal{L}(\mathbf{x}) = (\bar{\mathbf{w}} - \mathbf{w}^*)^T(\bar{\mathbf{e}}^T\bar{\mathbf{e}}H^2 + H\theta\theta^T H)(\bar{\mathbf{w}} - \mathbf{w}^*) \tag{27}$$

When comparing to the mean-model loss $\mathcal{L} = (\bar{\mathbf{w}} - \mathbf{w}^*)^T H (\bar{\mathbf{w}} - \mathbf{w}^*)$ we see that the two are tightly connected, both using a weighted distance measure between $\bar{\mathbf{w}}$ and $\mathbf{w}^*$, with only a different weighting. Taken everything together, we see that we can take the LHS of equation 23 (and hence also the RHS) as a rough proxy for the expected risk under the steady-state distribution (equation 5), which will be important to investigate the influence of the amount of ensemble members on the expected risk. Zhu et al. (2018) highlighted this trace quantitiy in equation 23 as a measurement of the escaping efficiency out of poor minima. However, we assume that we are in the final valley of convergence (emphasized by this convex NQP), so now this interpretation does not hold and the quantity should be considered as a proxy measurement of the width of the steady-state parameter distribution around the minimum. The trace quantity has $H_{\mathcal{L}}$ and $D(\mathbf{x}_{ss})^2$ as main elements, which we structure in block matrices below (for clarity and ease of notation, we drop the subscripts $ss$).

$$H_{\mathcal{L}} = \begin{bmatrix} (\bar{\mathbf{e}}\bar{\mathbf{e}}^T) \otimes H & \frac{1}{K}\mathbb{1}^T \otimes Q^T \\ \frac{1}{K}\mathbb{1} \otimes Q & \frac{1}{K^2}\mathbb{1} \otimes \theta^T H \theta \end{bmatrix} \tag{28}$$

$$D(\mathbf{x})^2 = G(I \otimes \Sigma)G^T = \begin{bmatrix} \frac{1}{K^2}(EE^T) \otimes (H\Sigma H) & \frac{1}{K}E \otimes (H\Sigma H\theta) \\ \frac{1}{K}E^T \otimes (\theta^T H\Sigma H) & I \otimes (\theta^T H\Sigma H\theta) \end{bmatrix} \tag{29}$$

with $\mathbb{1}$ a matrix or vector of the appropriate size full of ones, $\bar{\mathbf{e}} \triangleq 1/K \sum_k \mathbf{e}_k$ and the rows of $Q \in \mathbb{R}^{d \times nd}$ given by:

$$Q_{i,:} \triangleq \boldsymbol{\theta}^T \big((\bar{\mathbf{e}}\boldsymbol{\delta}_i^T + \boldsymbol{\delta}_i\bar{\mathbf{e}}^T) \otimes H\big) - \boldsymbol{\delta}_i^T \otimes (\mathbf{w}^{*T}H), \tag{30}$$

with $\boldsymbol{\delta}_i$ the $i$-th column of an appropriately sized identity matrix. After some intermediate calculations and rearranging of terms, we reach the following expression for the RHS of equation 23:

$$\frac{1}{2}\mathbb{E}_{\rho^{ss}}\big[\text{Tr}[\tilde{D}^2 H_{\mathcal{L}}]\big] = \frac{1}{K\beta}\Big(\mathbb{E}_{\rho^{ss}}\big[\text{Tr}[\tilde{E}^2\bar{\mathbf{e}}\bar{\mathbf{e}}^T]\big]\text{Tr}[H\Sigma H^2] + \mathbb{E}_{\rho^{ss}}\big[\text{Tr}[\bar{\mathbf{e}} \otimes (H\Sigma H\theta Q)] + ...$$

$$...\text{Tr}[(\bar{\mathbf{e}}^T \otimes (\theta^T H\Sigma H))Q^T)] + \text{Tr}[\theta^T H\Sigma H\theta\theta^T H\theta]\Big), \tag{31}$$

with $\tilde{E}^2 \triangleq \frac{1}{K}\sum_k \mathbf{e}_k\mathbf{e}_k^T = \frac{1}{K}EE^T$ Note that everything between the big brackets in the RHS is independent of $K$ in expectation. Hence, we see that the RHS of equation 23 scales inversely by $K$, exactly as the case for full ensembles (see Section C.3). Importantly, the approximation errors in equation 25 are independent of $K$, hence, the found scaling of $\frac{1}{K}$ in equation 31 translates to a scaling of $\frac{1}{K}$ of the expected risk of the NQP, following the above argumentation. Hence, we see that the non-trivial combination of (i) averaging over the base-weights gradients for all ensemble members and (ii) averaging the late-phase weights $\mathbf{e}_k$ in parameter space to obtain a mean-model, succeeds in scaling the expected loss after convergence inversely by $K$.

## C.3 NQP WITH FULL ENSEMBLES

As a comparison for the above theoretical results, we also analyze the NQP that uses an ensemble of $K$ full weight configurations $\mathbf{w}_k$ to get a mean model $\bar{\mathbf{w}}$, instead of shared weights $\theta$ and ensemble-member-specific weights $\phi_k$. For the case of linear models, the averaging in weight space to obtain a mean model is equivalent to the averaging of the predictions over the ensemble, which is conventionally done using ensembles. Without loss of generality, we can take $\mathbf{w}^* = 0$ (corresponding with a simple reparameterization of $\mathbf{w}$). Using equation 4, this results in the following parameter updates for the ensemble members:

$$\mathbf{w}_k^{(t+1)} = (I - \eta H)\mathbf{w}_k^{(t)} + \frac{\eta}{\sqrt{B}}H\boldsymbol{\epsilon}^{(t,k)} \tag{32}$$

The mean model $\bar{\mathbf{w}} \triangleq \frac{1}{K}\sum_k \mathbf{w}_k$ has the following corresponding discrete dynamics:

$$\bar{\mathbf{w}}^{(t+1)} = (I - \eta H)\bar{\mathbf{w}}^{(t)} + \frac{\eta}{K\sqrt{B}}H\sum_k \boldsymbol{\epsilon}^{(t,k)} \tag{33}$$

**Exact moment propagating equations.** As this is a discrete linear system with Gaussian noise, the resulting parameter distributions will also be linear and can be fully characterized by the mean

and covariance of the parameters. Taking the expectation and variance of equation 33 results in:

$$\mathbb{E}\big[\bar{\mathbf{w}}^{(t+1)}\big] = (I - \eta H)\mathbb{E}\big[\bar{\mathbf{w}}^{(t)}\big] \tag{34}$$

$$\mathbb{C}\big[\bar{\mathbf{w}}^{(t+1)}\big] = (I - \eta H)\mathbb{C}\big[\bar{\mathbf{w}}^{(t)}\big](I - \eta H) + \frac{\eta^2}{KB}H\Sigma H \tag{35}$$

with $\Sigma$ the covariance matrix of $\epsilon$. For an appropriate $\eta$, the above equations converge to the following fixed points at steady-state:

$$\mathbb{E}_{\rho^{ss}}\big[\bar{\mathbf{w}}\big] = \mathbf{0} \tag{36}$$

$$\mathrm{vec}\big(\mathbb{C}_{\rho^{ss}}\big[\bar{\mathbf{w}}\big]\big) = \frac{\eta^2}{KB}\big(I - (I - \eta H) \otimes (I - \eta H)\big)^{-1}\mathrm{vec}\big(H\Sigma H\big) \tag{37}$$

We see that the steady-state covariance of $\bar{\mathbf{w}}$ and hence of the risk $\mathcal{L}$ scales with $\frac{1}{K}$ ($\mathbb{E}_{\rho^{ss}}[\mathcal{L}] = \mathbb{E}_{\rho^{ss}}[\bar{\mathbf{w}}^T H \bar{\mathbf{w}}] = \mathrm{Tr}\big[H\mathbb{C}_{\rho^{ss}}[\bar{\mathbf{w}}]\big]$). The expected risk $\mathbb{E}_{\rho^{ss}}[\mathcal{L}]$ obtained with computationally expensive full ensembles can be seen as a lower limit that we try to reach with the *economical* ensembles of shared weights $\theta$ and late-phase weights $\phi_k$. Note that for the NQP, increasing the batchsize $B$ has a similar influence as increasing the number of ensemble members $K$, as can be seen in equation 37.

**Continuous time stochastic dynamics.** We can also do a similar continuous time analysis as Section C.2 for the case of full ensembles, to better relate it to the results of the late-phase learning with shared parameters. Following the same approach, we get the following expression for the trace term:

$$\frac{1}{2}\mathbb{E}_{\rho^{ss}}\big[\mathrm{Tr}[\tilde{D}^2 H_{\mathcal{L}}]\big] = \mathrm{Tr}\big[\frac{1}{\beta}\big(I \otimes (H\Sigma H)\big)\frac{1}{K^2}\big(\mathbb{1} \otimes H\big)\big] \tag{38}$$

$$= \frac{1}{K\beta}\mathrm{Tr}\big[H\Sigma H^2\big] \tag{39}$$

When comparing to equation 31, we see that the economical ensembles with shared parameters reach the same scaling with $\frac{1}{K}$ as a result of ensembling, however, some extra terms that vanish asymptotically for big $K$ appear as a result of the interplay between shared and late-phase parameters.

**Experimental details for Fig. 1.** We take the model $w = \theta \phi$ (i.e., $K = 1$) as our baseline, since this overparameterization could already result in accelerated learning (Arora et al., 2018). Our parameters are randomly initialized and scaled such that $\bar{w}$ has a fixed distance to $w^*$ of 1. Since the NQP mimics a late phase of learning we set $T_0 = 0$. We study a problem of dimension $n = 100$ and train the model with gradient descent (without momentum).

To validate the theoretical results, we show in Fig. 1 that the steady-state reached by our method scales inversely with $K$, similarly to an ensemble of independently-trained models. We run experiments with $K \in [2, 5, 10, 15, 20, 25]$ and train every configuration for $2 \times 10^7$ iterations until convergence. We average over the last $10^4$ weight updates and over 5 different random seeds.

