# OpenReview forum: "Neural networks with late-phase weights"
_ICLR.cc/2021/Conference — ICLR 2021 Poster_

### Official Review · AnonReviewer3 · 2020-10-27
**I think both the methodology and the writing need to be improved.**

**Rating:** 6
**Confidence:** 4

**Review:**

To improve the generalization performance of SGD methods,
this paper proposes to use an efficient ensemble-like approach
which computes an average of an ensemble of SGD weights
when retrained from some late-phase of SGD dynamics.
This idea is different to most recent ensemble-based approaches which
aim to average the predictions of the models.

The paper focuses on some specific layers of neural networks
in order to apply the late-phase training.
The batch normalization layers are shown to be
simple and effective. Some other layers are also analyzed,
including a recently introduced rank-1 multiplicative matrix
weights idea for full-connected layers.
Section 3 presents the numerical results and show that the generalization of SGD
is more-or-less improved on various benchmarks.
Explanation of why the generalization is improved in relation with the flatness
of energy landscape is also discussed.

I find that this approach is quite sensitive the choice of the hyper-parameters,
such as the beginning of the late-phase T0, and the noise perturbation sigma0.
It is written in Section 2.1 that in practice … sigma0>0 yields a set of models …
this results in improved final generalization. However, in the result of ImageNet in Section 3.3,
the sigma0 equals to 0. Thus, it is not conclusive that sigma0>0 is better.
As the improvement in Section 3.3 seems marginal compared to the baseline and the
standard deviation, it thus does not fully support the effectiveness of the batch normalization layers.
I would recommend using some other dataset or models,
but with a more consistent set of hyper-parameters.

In terms of writing, I would recommend to write out
the full algorithm of Alg. 1 or at least in the Appendix,
including the variant of the SGD momentum and Adam.
The SWA is also worth writing out clearly, which is not clear to the reader.
Is the DeepEnsemble result in Table 1 from SGD or SWA?
This is not clear from the text.

Overall, I think both the methodology and the writing need to be improved.

##
The revisions made by the authors have addressed all my concerns.

---

> ### Author Response · Authors · 2020-11-19
> **Reply to AnonReviewer3 (1/2)**
>
> Thank you for your review. As detailed below, we have tried to address your criticism to the best of our knowledge, and we remain open to any questions that you may have, that can help you raise your score.
>
> * *I find that this approach is quite sensitive the choice of the hyper-parameters,  such as the beginning of the late-phase T0, and the noise perturbation sigma0.  It is written in Section 2.1 that in practice … sigma0>0 yields a set of models … this results in improved final generalization. However, in the result of ImageNet in Section 3.3,  the sigma0 equals to 0. Thus, it is not conclusive that sigma0>0 is better.*
>
> Thank you for this comment. We recognize the seeming inconsistency of our choice of  $\sigma_0$, and now present our results on CIFAR and ImageNet in the main text all using a consistent choice of $\sigma_0 = 0$ (and $T_0$, for non-pretrained models). A non-zero $\sigma_0 = 0.5$ is now employed only to generate a diverse late-phase ensemble (see Section 3.2, Table 4) that is not averaged in weight space. This method shows strong OOD performance in comparison to other comparable techniques that are efficiently-trained but still require to integrate predictions during inference (SWAG, MC-dropout, BatchEnsemble). An overview of the updated performance can be found in the table below.
>
> \
> +-------------------------------------------------------------------+--------------------+---------------------+-------------------+\
> |                       |Testacc.(%)	 |&nbsp;&nbsp; OOD  	 | 	mCE   	|\
> +-------------------------------------------------------------------+--------------------+---------------------+-------------------+\
> | Base(SGD)                 |81.35+/-0.16 |0.802+/-0.019 |47.84+/-0.41 |\
> | Dropout(Mean)(SGD)            |81.31+/-0.20 |0.802+/-0.030 |48.97+/-0.33 |\
> | Late-phase BatchNorm(SGD)&nbsp;&nbsp;&nbsp;&nbsp;&nbsp;&nbsp;&nbsp;&nbsp;
>    |82.87+/-0.14 |0.836+/-0.012 |45.59+/-0.25 |\
> |                       |&nbsp;&nbsp;      |&nbsp;&nbsp;       |&nbsp;&nbsp;      |\
> | MC-Dropout(SGD)&nbsp;&nbsp;             |81.55+/-0.11 |0.823+/-0.049 |48.09+/-0.36 |\
> | SWAG(SWA)                &nbsp;|82.12+/-0.03 |0.828+/-0.027 |&nbsp;&nbsp;      |\
> | BatchEnsemble(SGD)            |81.25+/-0.10 |0.829+/-0.019 |&nbsp;&nbsp;      |\
> | Late-phase BatchNorm(SGD,non-averaged)&nbsp;&nbsp;&nbsp;&nbsp;&nbsp;&nbsp;|82.71+/-0.10 |0.862+/-0.009 |46.21+/-0.29 |\
> |                       |&nbsp;&nbsp;      |&nbsp;&nbsp;       |&nbsp;&nbsp;      |\
> | Deepensemble (SGD)&nbsp;&nbsp;            |84.09&nbsp;    |0.8312&nbsp;    |44.21    	|\
> | Deepensemble (Late-phaseBatchNorm,SGD)&nbsp;&nbsp;&nbsp;&nbsp;|84.69&nbsp;    |0.8575&nbsp;    |43.15    	|\
> +-------------------------------------------------------------------+--------------------+---------------------+-------------------+
>
>
> For more details on hyperparameter sensitivity, we would like to refer to our analyses (Table 12 and Figure 6) supporting that our method is robust to the choice of $T_0$. Here we show that for a large range of $T_0$, i.e., $T_0 > 80$, we improve on top of the baseline (CIFAR-10 - 96.16% and CIFAR-100 - 81.31%) in test set accuracy and out-of-distribution detection. See a small excerpt of the analyses here:
>
> +---------+------------------------+---------------------+ \
> |&nbsp;&nbsp;&nbsp;&nbsp;T_0&nbsp;&nbsp;|&nbsp;&nbsp;&nbsp;&nbsp;&nbsp;&nbsp;&nbsp;CIFAR10&nbsp;&nbsp;&nbsp;&nbsp;&nbsp;&nbsp;&nbsp;&nbsp;&nbsp;|&nbsp;&nbsp;&nbsp;&nbsp;&nbsp;&nbsp;CIFAR100&nbsp;&nbsp;&nbsp;&nbsp;&nbsp;|&nbsp;\
> +---------+------------------------+---------------------+\
> |&nbsp;&nbsp;&nbsp;&nbsp;40&nbsp;&nbsp;&nbsp;&nbsp;|&nbsp;&nbsp;&nbsp;96.34&nbsp;+/-&nbsp;0.08&nbsp;&nbsp;&nbsp;|&nbsp;&nbsp;79.69&nbsp;+/-&nbsp;0.11&nbsp;&nbsp;|\
> |&nbsp;&nbsp;&nbsp;&nbsp;80&nbsp;&nbsp;&nbsp;&nbsp;|&nbsp;&nbsp;&nbsp;96.50&nbsp;+/-&nbsp;0.11&nbsp;&nbsp;&nbsp;|&nbsp;&nbsp;81.72&nbsp;+/-&nbsp;0.18&nbsp;&nbsp;|\
> |&nbsp;&nbsp;&nbsp;100&nbsp;&nbsp;&nbsp;|&nbsp;&nbsp;&nbsp;96.45&nbsp;+/-&nbsp;0.08&nbsp;&nbsp;&nbsp;|&nbsp;&nbsp;82.48&nbsp;+/-&nbsp;0.21&nbsp;&nbsp;|\
> |&nbsp;&nbsp;&nbsp;120&nbsp;&nbsp;&nbsp;|&nbsp;&nbsp;&nbsp;96.48&nbsp;+/-&nbsp;0.20&nbsp;&nbsp;&nbsp;|&nbsp;&nbsp;82.87&nbsp;+/-&nbsp;0.18&nbsp;&nbsp;|\
> |&nbsp;&nbsp;&nbsp;140&nbsp;&nbsp;&nbsp;|&nbsp;&nbsp;&nbsp;96.26&nbsp;+/-&nbsp;0.17&nbsp;&nbsp;&nbsp;|&nbsp;&nbsp;82.53&nbsp;+/-&nbsp;0.21&nbsp;&nbsp;|\
> |&nbsp;&nbsp;&nbsp;160&nbsp;&nbsp;&nbsp;|&nbsp;&nbsp;&nbsp;96.23&nbsp;+/-&nbsp;0.11&nbsp;&nbsp;&nbsp;|&nbsp;&nbsp;81.41&nbsp;+/-&nbsp;0.31&nbsp;&nbsp;|\
> +----------+-----------------------+----------------------+

---

> > ### Author Response · Authors · 2020-11-19
> > **Reply to AnonReviewer3 (2/2)**
> >
> > * *As the improvement in Section 3.3 seems marginal compared to the baseline and the  standard deviation, it thus does not fully support the effectiveness of the batch normalization layers.*
> >
> > Thank you for your assessment. We have now updated and expanded Section 3.3 with additional models which better showcase the effectiveness of late-phase batch normalization. The reviewer can find some of the new results in the table below, and in Table 5. We now show significant gains in accuracy (up to 0.4%), especially when considering that we are fine-tuning an off-the-shelf pretrained network.
> >
> > +---------------+--------------------------+-------------------+--------------------+\
> > |  &nbsp;Dataset    &nbsp;&nbsp; | &nbsp;&nbsp;&nbsp; &nbsp;&nbsp; &nbsp;&nbsp;     Model       &nbsp;&nbsp; &nbsp;&nbsp;&nbsp;&nbsp; &nbsp;&nbsp;            |      &nbsp;&nbsp; &nbsp;  Base &nbsp;&nbsp; &nbsp;&nbsp; &nbsp;&nbsp;          |   &nbsp;&nbsp;Late-phase &nbsp;   |\
> > +---------------+--------------------------+-------------------+--------------------+\
> > | ImageNet&nbsp;| &nbsp;&nbsp;&nbsp;&nbsp;&nbsp;&nbsp; ResNet-152&nbsp;&nbsp;&nbsp;&nbsp; | 78.37 +/- 0.01 | 78.77 +/- 0.01  |\
> > | CIFAR-10 &nbsp; | WRN28-14 (SWA)&nbsp;  | 96.75 +/- 0.05 | 97.45 +/- 0.10  |\
> > | CIFAR-100&nbsp;| WRN28-14 (SWA)&nbsp;  | 84.01 +/- 0.29 | 85.00 +/- 0.25  |\
> > +---------------+--------------------------+-------------------+--------------------+
> >
> > We showcase the efficiency of late-phase BatchNorm weights in fine-tuning ImageNet, but we see it evidently on full training. In particular, on a WRN 28-14 (Table 3), we increase accuracy from 96.75% (SWA) to 97.45% (Late-phase+SWA) on CIFAR-10, and from 84.01% (SWA) to 85.00% (Late-phase+SWA) on CIFAR-100. Both are very high accuracies for ResNet models (in PWC [1, 2] this would be the best published result for CIFAR-100; on CIFAR-10 we would be second.).
> >
> > * *In terms of writing, I would recommend to write out the full algorithm of Alg. 1 or at least in the Appendix,  including the variant of the SGD momentum and Adam.  The SWA is also worth writing out clearly, which is not clear to the reader.*
> >
> > Done, this is a good suggestion, in particular in regards to SWA, as the paper did not stand on its own.  We now present lower-level pseudocode in Appendix A (Figures 4 and 5).
> >
> > * *Overall, I think both the methodology and the writing need to be improved.*
> >
> > As you can see from our answers to the other reviewers we have taken several steps to improve the clarity, readability and methodology of the revised paper. If writing and methodology are still a concern it would help us if you could point more specifically to where you think further improvements are required.
> >
> > Finally, we welcome the reviewer to take a second look into the paper and to possibly reassess his rather low rating with a particular focus on our revised text and the stronger experimental section. We remain open to any criticism and feedback on parts that could still be improved.
> >
> > [1] https://paperswithcode.com/sota/image-classification-on-cifar-100 \
> > [2] https://paperswithcode.com/sota/image-classification-on-cifar-10

---

> > > ### Comment · AnonReviewer3 · 2020-11-21
> > > **concern about reproducibility**
> > >
> > > Thanks for the revision. I still have some concern regarding the late-phase training algorithm, as mentioned in Alg 4 of Fig 5 (on  page 16), what is the stoppling criteria of the method (i.e. in the while loop, what does it mean not converged)? If all the main results could be made reproducible, I'd be happy to raise my score.

---

> > > > ### Author Response · Authors · 2020-11-22
> > > > **Code added as SM**
> > > >
> > > > Thank you for your quick response. We have now clarified Algorithm 4: the main loop runs while $t \leq T$, where $T$ is the total number of minibatches of data consumed. We would like to stress that, in all our experiments, our algorithm passes the same number of times through the dataset as standard optimization of a single model would do. We do not train our late-phase weights for any longer (see also Table 17 for an absolute runtime comparison in seconds).
> > > > Furthermore, we have attached code to reproduce our main WRN 28-10, WRN 28-14 (predictive test set and aggregate OOD scores) and ImageNet results as a supplementary material zip. Code for reproducing the remaining experiments will be uploaded with the final version of the paper.
> > > >
> > > > Do not hesitate in contacting us if you have any additional questions.

---

### Official Review · AnonReviewer4 · 2020-10-28

**Rating:** 7
**Confidence:** 4

**Review:**

This work suggests a variant of ensembling that is more compute-efficient. Specifically, it involves forking an ensemble only in the late stage of training, and forming this ensemble via a "low-dimentional" family. That is, instead of maintaining independent networks, maintain only "low-rank"-style perturbations of the base network (for various instanciations of "low-rank").
The experimental results are somewhat limited, but appear to be competitive with current efficient-ensembling approaches like SWA/SWAG. The absolute improvement of this method is not very large (<0.3% on CIFAR, <0.2% on imagenet), and there is a large gap to Deep Ensembles. I weakly recommend acceptance, because the method appears promising for future work, and the experiments seem correct.

There is also a theory section included, though I am generally unconvinced by results in such simple toy examples.
(such settings can usually be contrived to exhibit any desired behavior)

Weaknesses:
- The experimental section would be greatly strengthened by additional experiments for different models and settings. There are only 2 architectures tested on CIFAR-10, for example. It would also be informative to see the performance of these methods in "harder" settings -- for example, CIFAR-10 with fewer train samples.
- The OOD uncertainty results could be expanded. Uncertainty estimation and robustness are some of the most relevant practical uses of ensemble methods, so it is especially important to evaluate ensembles in this context. Currently aggregate results are shown in Table 4, but it would be good to explicitly see, for example: how the performance of this method degrades with increasing CIFAR-10C corruption severity, as opposed to Deep Ensembles. Also, reporting the Mean Corruption Error (mCE) for each dataset individually will allow standard comparison to prior methods.

Comments which do not affect the score:
It seems that starting the ensembling at a "late phase" in training is the main contribution of this work. This could be applied to any ensemble method, and you propose several explicit instantiations. It could help to focus the writing in terms of this contribution, and also to further investigate the role of T0 (the time at which ensembling starts).

---
Edit after rebuttal: Increased score from 6 to 7.

---

> ### Author Response · Authors · 2020-11-19
> **Reply to AnonReviewer4 (1/2)**
>
> Thank you for your thorough review and constructive criticism. We have followed your specific suggestions and expanded our OOD experiments, enlarged our coverage of models, and considered alternative use cases involving ensembling the solutions found with our method. We reply point-to-point below:
>
> * *The experimental results are somewhat limited, but appear to be competitive with current efficient-ensembling approaches like SWA/SWAG. The absolute improvement of this method is not very large (<0.3% on CIFAR, <0.2% on imagenet), and there is a large gap to Deep Ensembles. [...] The experimental section would be greatly strengthened by additional experiments for different models and settings.*
>
>
> We have now extended our experiments to include efficient-ensembling baselines (dropout, MC-dropout, BatchEnsemble) and new network architectures showing that our results are in fact strong compared to the performance increases achieved with other methods.
> Below, we highlight our current results for larger models (test set accuracy shown in %):
>
> +---------------+--------------------------+-------------------+--------------------+\
> |  &nbsp;Dataset    &nbsp;&nbsp; | &nbsp;&nbsp;&nbsp; &nbsp;&nbsp; &nbsp;&nbsp;     Model       &nbsp;&nbsp; &nbsp;&nbsp;&nbsp;&nbsp; &nbsp;&nbsp;            |      &nbsp;&nbsp; &nbsp;  Base &nbsp;&nbsp; &nbsp;&nbsp; &nbsp;&nbsp;          |   &nbsp;&nbsp;Late-phase &nbsp;   |\
> +---------------+--------------------------+-------------------+--------------------+\
> | ImageNet&nbsp;| &nbsp;&nbsp;&nbsp;&nbsp;&nbsp;&nbsp; ResNet-152&nbsp;&nbsp;&nbsp;&nbsp; | 78.37 +/- 0.01 | 78.77 +/- 0.01  |\
> | CIFAR-10 &nbsp; | WRN28-14 (SWA)&nbsp;  | 96.75 +/- 0.05 | 97.45 +/- 0.10  |\
> | CIFAR-100&nbsp;| WRN28-14 (SWA)&nbsp;  | 84.01 +/- 0.29 | 85.00 +/- 0.25  |\
> +---------------+--------------------------+-------------------+--------------------+
>
> These WRN 28-14 gains, obtained on top of SWA, place us among the best results for WRNs reported in PWC [1, 2]. The improvement on ImageNet is significant for a fine-tuning method.
>
> Further, we would like to highlight that since our method yields a single model (unlike MC-dropout or BatchEnsemble), it can be used to obtain a stronger DeepEnsemble in a straightforward manner. To showcase this we now present a proof-of-concept experiment showing that this is another possible use case, if one has the resources to build a DeepEnsemble (CIFAR-10 in Table 1 & CIFAR-100 in Table 4).
>
> * *I weakly recommend acceptance, because the method appears promising for future work, and the experiments seem correct.*
>
> Thank you for the encouraging feedback.  In light of your and the other reviewers’ comments we have performed new experiments and added new baselines to our work (see general comment). This has significantly  strengthened the paper and we hope that you now find it worthy of a ‘clear acceptance’.
>
> * *There is also a theory section included, though I am generally unconvinced by results in such simple toy examples. (such settings can usually be contrived to exhibit any desired behavior).*
>
> Thank you for this comment -- we have rewritten Section 3.1, to clarify and discuss the main contributions and implications of our new theory. We agree that going beyond the NQP would be desirable, but with the current analytical tools that is likely out-of-reach and beyond the scope of this work.
>
> * *There are only 2 architectures tested on CIFAR-10, for example. It would also be informative to see the performance of these methods in "harder" settings -- for example, CIFAR-10 with fewer train samples.*
>
> We thank you for this comment and now include results when training with less (a fifth) CIFAR-10 examples (Table 15). These complement our new results obtained on a larger WRN (the WRN 28-14, Table 3 and table above) on CIFAR-10. In both cases, our implicit regularization leads to performance gains (respectively +0.6%, +0.7%) that are higher than on the WRN 28-10 (+0.3%).
>
> * *The OOD uncertainty results could be expanded. Uncertainty estimation and robustness are some of the most relevant practical uses of ensemble methods, so it is especially important to evaluate ensembles in this context.*
>
> We fully agree and have expanded our OOD section in response to your comment. We have improved our scores and considered yet another use case for our model: maintaining our late-phase ensemble at the end of training. Like SWAG, MC-dropout and BatchEnsemble, in this case, we integrate predictions across an efficiently-trained ensemble. This resulted in strongly improved OOD scores for our method (Table 4).

---

> > ### Author Response · Authors · 2020-11-19
> > **Reply to AnonReviewer4 (2/2)**
> >
> > * *Currently aggregate results are shown in Table 4, but it would be good to explicitly see, for example: how the performance of this method degrades with increasing CIFAR-10C corruption severity, as opposed to Deep Ensembles. Also, reporting the Mean Corruption Error (mCE) for each dataset individually will allow standard comparison to prior methods.*
> >
> > Thank you. As suggested, we computed the mCE on CIFAR-100-C:
> >
> > \
> > +-------------------------------------------------------------------+--------------------+---------------------+-------------------+\
> > |                       |Testacc.(%)	 |&nbsp;&nbsp; OOD  	 | 	mCE   	|\
> > +-------------------------------------------------------------------+--------------------+---------------------+-------------------+\
> > | Base(SGD)                 |81.35+/-0.16 |0.802+/-0.019 |47.84+/-0.41 |\
> > | Dropout(Mean)(SGD)            |81.31+/-0.20 |0.802+/-0.030 |48.97+/-0.33 |\
> > | Late-phase BatchNorm(SGD)&nbsp;&nbsp;&nbsp;&nbsp;&nbsp;&nbsp;&nbsp;&nbsp;
> >    |82.87+/-0.14 |0.836+/-0.012 |45.59+/-0.25 |\
> > |                       |&nbsp;&nbsp;      |&nbsp;&nbsp;       |&nbsp;&nbsp;      |\
> > | MC-Dropout(SGD)&nbsp;&nbsp;             |81.55+/-0.11 |0.823+/-0.049 |48.09+/-0.36 |\
> > | SWAG(SWA)                &nbsp;|82.12+/-0.03 |0.828+/-0.027 |&nbsp;&nbsp;      |\
> > | BatchEnsemble(SGD)            |81.25+/-0.10 |0.829+/-0.019 |&nbsp;&nbsp;      |\
> > | Late-phase BatchNorm(SGD,non-averaged)&nbsp;&nbsp;&nbsp;&nbsp;&nbsp;&nbsp;|82.71+/-0.10 |0.862+/-0.009 |46.21+/-0.29 |\
> > |                       |&nbsp;&nbsp;      |&nbsp;&nbsp;       |&nbsp;&nbsp;      |\
> > | Deepensemble (SGD)&nbsp;&nbsp;            |84.09&nbsp;    |0.8312&nbsp;    |44.21    	|\
> > | Deepensemble (Late-phaseBatchNorm,SGD)&nbsp;&nbsp;&nbsp;&nbsp;|84.69&nbsp;    |0.8575&nbsp;    |43.15    	|\
> > +-------------------------------------------------------------------+--------------------+---------------------+-------------------+
> >
> > The full table can be found in the Appendix B, together with all non-aggregate OOD results (Table 16).
> >
> > * *It seems that starting the ensembling at a "late phase" in training is the main contribution of this work. This could be applied to any ensemble method, and you propose several explicit instantiations.*
> >
> > Following this viewpoint, we now show that skipping the final weight averaging step and maintaining the late-phase-initiated ensemble at the end of training can be beneficial in OOD problems, as described above.
> >
> > In addition, aligned with this view and following AnonReviewer1’s suggestion, we also ran a new experiment using a (non-memory-efficient) full DeepEnsemble created in a late-phase (reported inline in Section 3.2) and trained with the same data consumption as a single model. This did not improve performance as strongly as our multiplicative late-phase models, highlighting the importance of an appropriately-chosen low-dimensional set of late-phase weights.
> >
> > * *[...] to further investigate the role of T0 (the time at which ensembling starts)*
> >
> > In the revised version of the paper, we now present in Table 12 and Figure 5 an extended analysis on $T_0$, which is indeed the main hyperparameter of the algorithm; in particular, it should not be set too early (which motivated the late-phase term in our algorithm).
> >
> > To summarize, we have addressed all major concerns and if you agree that our paper has significantly improved we would be grateful for a reassessment of the work and rating. We are happy to answer any further questions that you would like to see addressed.

---

> > > ### Comment · AnonReviewer4 · 2020-11-19
> > > **response to authors**
> > >
> > > Thank you for the follow-up experiments and revised writing.
> > > I will increase my score to '7', since the experimental section is now thorough enough that I believe it provides value to the community.
> > >
> > > Although the absolute improvement is not a breakthrough, it is hard to make large progress in this field, and I believe this paper contributes to our understanding of the impact of ensembling (and "late-phase" ensembling).
> > > I believe the experiments present in this paper will save time for future researchers interested in similar questions (since they are now more extensive, and compared to existing methods), and this could be built upon in future work.

---

> > > > ### Author Response · Authors · 2020-11-20
> > > > **Thank you**
> > > >
> > > > Thank you for your prompt reassessment of our work, we very much appreciate it.

---

### Official Review · AnonReviewer2 · 2020-10-28
**Official Blind Review #2**

**Rating:** 6
**Confidence:** 2

**Review:**

Summary:

The paper proposes a method to improve solutions found by SGD by ensembling subsets of weights in late-phase. A family of low-dimensional late-phase methods are analyzed and shown to improve generalization in CIFAR-10/100, ImageNet and enwik8. Authors also analyze the method in more tractable noisy quadratic settings.

Contribution of the authors is that rather obtaining ensemble they utilize efficient ensemble to guide SGD training and ultimately obtain a single model.

Reason for score:

While the paper discusses efficient ways of utilizing late-phase weight ensemble and improving SGD training, the demonstrated benefit is not significant enough for practitioners to pursue the method. Without strong practical application potential, merit of the proposed method is weak since it does not obviously elucidate some aspects of neural network training.

Pros:

The paper is clearly written and easy to understand the proposed method is. It is well structured that helps to improve the clarity.

Proposed method tackles a significant problem in the standard ensemble method in which both training/inference computation can be quite costly. The paper’s method only ensembles subset of weights therefore added training cost is minimal and since inference is done on averaged weight, it becomes essentially a single model.

Among various late-phase schemes, BatchNorm late-phase seems to work well which is widely used among vision models so easily applicable. Also since late-phase can be applied post-pretraining, it can be used to improve pre-trained models.

As far as I can tell various experimental conditions are very well controlled and thoughtfully designed.

Cons:
The idea of weight averaging is not so novel as duly noted by the authors.

Main question arises for the paper is whether the proposed method is worth the effort. While all experiments show that the proposed method improves the baseline somewhat, deep ensemble baselines remain strong. Also quoted difference between methods does not mean statistically significant effect (see Vincent Vanhoucke’s article on reporting significant figures https://towardsdatascience.com/digit-significance-in-machine-learning-dea05dd6b85b). According to this article, results reported in Table 1, CIFAR-10 in WRN, a significant figure with a 10k test set should be around 0.2% and differences between different methods are at best marginal. This can be applied to most tables and except for Deep Ensemble’s improvement other differences are not very significant.

I wonder as discussed by the authors, this is due to mostly the benefit of ensembles is
through incorporating different modes as argued in [Fort et al., 2020] rather than a single mode. I imagine a single mode ensemble could be beneficial when variance within the mode is large, however for models considered by the authors seem to have small model variance which minimizes effect of technique utilizing single mode.

While \sigma_0 and T_0 are hyperparameters of the algorithm, no good way to determine it is explained.

The role of section 3.1 is not clear. For one thing, the legend in Figure 1 is confusing where the role of non-integer K is mysterious to me. I would suggest clarifying what the message of the section would be in context of understanding late-phase weight models.


Nits and additional feedback:

Anonymized link is neither there in the main paper or included as supplementary material. If the authors intended to include the code, this is a note that code can not be found to the reviewers.

For models that do not use BatchNorm, I believe most interest to practitioners would be using Transformer based models. I wonder if rank-1 late-phase or LayerNorm late-phase would show improvements in this case.

Was “Late-phase classification layers” ever evaluated or discussed in the main paper? I find some discussion on the appendix but seem to be missing in the main text.

---
I thank the authors for their hard work addressing issues raised by the reviewers.

Authors have answered many issues pointed out (by improved performance and showing robustness to hyperparameters) and I've increased my score from 5 to 6, and support accepting the paper.

---

> ### Author Response · Authors · 2020-11-19
> **Reply to AnonReviewer2 (1/2)**
>
> Thank you for your thorough review and feedback. We ran a number of new experiments, which include an extended set of OOD results, the study of more network architectures, and new well-known baselines (dropout, MC-dropout, and BatchEnsemble) with comparable computational and memory requirements to our method. We reply to your comments point by point:
>
> * *BatchNorm late-phase seems to work well which is widely used among vision models so easily applicable. Also since late-phase can be applied post-pretraining, it can be used to improve pre-trained models.*
>
> To facilitate broad adoption of our method we will provide a PyTorch drop-in replacement for a standard BatchNorm layer with the final version of the paper. Our aim is to make our method as easy to implement as other well-established complementary techniques like dropout or SWA.
>
> * *The idea of weight averaging is not so novel as duly noted by the authors.*
>
> While previous optimization algorithms employ various sorts of weight averaging (notably, SWA and Polyak averaging, which maintain running temporal averages), we would like to highlight that our approach differs in that we take a single, simple spatial average in a low-dimensional weight space. This simple averaging is made possible thanks to the late-phase ensembling that we introduce in our paper, as corroborated by our experiments where $T_0$ is varied (Table 12, Figure 5).
> * *While the paper discusses efficient ways of utilizing late-phase weight ensemble and improving SGD training, the demonstrated benefit is not significant enough for practitioners to pursue the method. Without strong practical application potential, merit of the proposed method is weak since it does not obviously elucidate some aspects of neural network training.
> ….
> Main question arises for the paper is whether the proposed method is worth the effort. While all experiments show that the proposed method improves the baseline somewhat, deep ensemble baselines remain strong. Also quoted difference between methods does not mean statistically significant effect [...]. Results reported in Table 1, CIFAR-10 in WRN, a significant figure with a 10k test set should be around 0.2% and differences between different methods are at best marginal. This can be applied to most tables and except for Deep Ensemble’s improvement other differences are not very significant.*
>
> To convince you that our paper is worth accepting we would like to point out our improved results (Tables 2, 3, 4 and 5 in the main text). Here, we would like to highlight that on a WRN 28-14 we increase accuracy from 96.75% (SWA) to 97.45% (Late-phase+SWA) on CIFAR-10, and from 84.01% (SWA) to 85.00% (Late-phase+SWA) on CIFAR-100. These high accuracies place us among the very best available results for WRNs in PWC [1, 2] (1st place on CIFAR-100 and 2nd place on CIFAR-10). Furthermore, these results also show our method and SWA, one of the strongest methods for improving generalization in neural networks, are complementary.
>
> On the point raised over statistical significance, we stress that our CIFAR results are obtained with a consistent choice of $T_0$ and $\sigma_0$, across a number of different architectures.
>
> Our updated results also show that our method achieves strong performance in out-of-distribution detection problems and it is robust to input data corruptions: (Table 4 and 16 in the paper).
> \
> +-------------------------------------------------------------------+--------------------+---------------------+-------------------+\
> |                       |Testacc.(%)	 |&nbsp;&nbsp; OOD  	 | 	mCE   	|\
> +-------------------------------------------------------------------+--------------------+---------------------+-------------------+\
> | Base(SGD)                 |81.35+/-0.16 |0.802+/-0.019 |47.84+/-0.41 |\
> | Dropout(Mean)(SGD)            |81.31+/-0.20 |0.802+/-0.030 |48.97+/-0.33 |\
> | Late-phase BatchNorm(SGD)&nbsp;&nbsp;&nbsp;&nbsp;&nbsp;&nbsp;&nbsp;&nbsp;
>    |82.87+/-0.14 |0.836+/-0.012 |45.59+/-0.25 |\
> |                       |&nbsp;&nbsp;      |&nbsp;&nbsp;       |&nbsp;&nbsp;      |\
> | MC-Dropout(SGD)&nbsp;&nbsp;             |81.55+/-0.11 |0.823+/-0.049 |48.09+/-0.36 |\
> | SWAG(SWA)                &nbsp;|82.12+/-0.03 |0.828+/-0.027 |&nbsp;&nbsp;      |\
> | BatchEnsemble(SGD)            |81.25+/-0.10 |0.829+/-0.019 |&nbsp;&nbsp;      |\
> | Late-phase BatchNorm(SGD,non-averaged)&nbsp;&nbsp;&nbsp;&nbsp;&nbsp;&nbsp;|82.71+/-0.10 |0.862+/-0.009 |46.21+/-0.29 |\
> |                       |&nbsp;&nbsp;      |&nbsp;&nbsp;       |&nbsp;&nbsp;      |\
> | Deepensemble (SGD)&nbsp;&nbsp;            |84.09&nbsp;    |0.8312&nbsp;    |44.21    	|\
> | Deepensemble (Late-phaseBatchNorm,SGD)&nbsp;&nbsp;&nbsp;&nbsp;|84.69&nbsp;    |0.8575&nbsp;    |43.15    	|\
> +-------------------------------------------------------------------+--------------------+---------------------+-------------------+

---

> > ### Author Response · Authors · 2020-11-19
> > **Reply to AnonReviewer2 (2/2)**
> >
> > * *I wonder as discussed by the authors, this is due to mostly the benefit of ensembles is through incorporating different modes as argued in [Fort et al., 2020] rather than a single mode.*
> >
> > Yes; we now provide improved OOD results using a late-phase weight ensemble obtained with large initialization noise, where weight averaging fails (Table 4), lending further credibility to this hypothesis.
> >
> > * *While $\sigma_0 $ and $T_0$ are hyperparameters of the algorithm, no good way to determine it is explained.*
> >
> > To get best results these hyperparameters should be indeed tuned on a validation set. We note that this equally applies to e.g. the dropout probability (incidentally, we also present a supplementary analysis for the sensitivity of dropout to its hyperparameter, Table 13).
> >
> > We therefore performed a finer hyperparameter scan showing that good results can be achieved when tuning only $T_0$, and that the range of optimal $T_0$ is not overly narrow (cf. Table 12, Figure 5). Please take into consideration the new robustness analyses of $T_0$ in Table 12 and Figure 6 in the appendix. Here we show, that for a large range of $T_0$, i.e. $T_0 > 80$, we improve on top of the baseline (CIFAR-10 - 96.16% and CIFAR-100 - 81.31%) in test set accuracy and out-of-distribution detection. We provide a small excerpt of the analyses below:
> >
> > +---------+------------------------+---------------------+ \
> > |&nbsp;&nbsp;&nbsp;&nbsp;T_0&nbsp;&nbsp;|&nbsp;&nbsp;&nbsp;&nbsp;&nbsp;&nbsp;&nbsp;CIFAR10&nbsp;&nbsp;&nbsp;&nbsp;&nbsp;&nbsp;&nbsp;&nbsp;&nbsp;|&nbsp;&nbsp;&nbsp;&nbsp;&nbsp;&nbsp;CIFAR100&nbsp;&nbsp;&nbsp;&nbsp;&nbsp;|&nbsp;\
> > +---------+------------------------+---------------------+\
> > |&nbsp;&nbsp;&nbsp;&nbsp;40&nbsp;&nbsp;&nbsp;&nbsp;|&nbsp;&nbsp;&nbsp;96.34&nbsp;+/-&nbsp;0.08&nbsp;&nbsp;&nbsp;|&nbsp;&nbsp;79.69&nbsp;+/-&nbsp;0.11&nbsp;&nbsp;|\
> > |&nbsp;&nbsp;&nbsp;&nbsp;80&nbsp;&nbsp;&nbsp;&nbsp;|&nbsp;&nbsp;&nbsp;96.50&nbsp;+/-&nbsp;0.11&nbsp;&nbsp;&nbsp;|&nbsp;&nbsp;81.72&nbsp;+/-&nbsp;0.18&nbsp;&nbsp;|\
> > |&nbsp;&nbsp;&nbsp;100&nbsp;&nbsp;&nbsp;|&nbsp;&nbsp;&nbsp;96.45&nbsp;+/-&nbsp;0.08&nbsp;&nbsp;&nbsp;|&nbsp;&nbsp;82.48&nbsp;+/-&nbsp;0.21&nbsp;&nbsp;|\
> > |&nbsp;&nbsp;&nbsp;120&nbsp;&nbsp;&nbsp;|&nbsp;&nbsp;&nbsp;96.48&nbsp;+/-&nbsp;0.20&nbsp;&nbsp;&nbsp;|&nbsp;&nbsp;82.87&nbsp;+/-&nbsp;0.18&nbsp;&nbsp;|\
> > |&nbsp;&nbsp;&nbsp;140&nbsp;&nbsp;&nbsp;|&nbsp;&nbsp;&nbsp;96.26&nbsp;+/-&nbsp;0.17&nbsp;&nbsp;&nbsp;|&nbsp;&nbsp;82.53&nbsp;+/-&nbsp;0.21&nbsp;&nbsp;|\
> > |&nbsp;&nbsp;&nbsp;160&nbsp;&nbsp;&nbsp;|&nbsp;&nbsp;&nbsp;96.23&nbsp;+/-&nbsp;0.11&nbsp;&nbsp;&nbsp;|&nbsp;&nbsp;81.41&nbsp;+/-&nbsp;0.31&nbsp;&nbsp;|\
> > +----------+-----------------------+----------------------+
> >
> >
> >
> > * *The role of section 3.1 is not clear. For one thing, the legend in Figure 1 is confusing where the role of non-integer K is mysterious to me. I would suggest clarifying what the message of the section would be in context of understanding late-phase weight models.*
> >
> > We followed the reviewer’s suggestion and rewrote this section, discussing directly in the main text our analytical results, as analytical tractability was the primary motivation for studying the noisy quadratic problem. In addition, we replaced the figure by a new one which directly validates the theoretical claims (Figure 1). The previous legend was indeed unclear (the non-integer number referred to the learning rate).
> >
> > * *Was “Late-phase classification layers” ever evaluated or discussed in the main paper? I find some discussion on the appendix but seem to be missing in the main text.*
> >
> > They were presented as the last type of late-phase weight models (in Section 2.2). In addition, we now mention their use again on the results subsection.
> >
> > In summary, we have verified the hyperparameter robustness of our approach, we now provide stronger performance results and we improved the clarity of the revised paper. If you agree with this improvement we would welcome any reassessment of our work and the rather low rating. We remain open to discuss and will address any remaining concerns regarding your assessment of our work.
> >
> > [1] https://paperswithcode.com/sota/image-classification-on-cifar-100 \
> > [2] https://paperswithcode.com/sota/image-classification-on-cifar-10

---

### Official Review · AnonReviewer1 · 2020-10-29

**Rating:** 7
**Confidence:** 4

**Review:**


### Summary
The authors propose late-phase weights, a method of updating the weights near the end of training via a splitting and ensembling mechanism. They analyze the benefits in the noisy quadratic setting. The method improves validation performance on a range of image recognition tasks and on enwiki8.

### Comments
* The weight interaction functions $h$ should be more explicitly defined rather than just described in text.
* The paper is overall well written and flows smoothly.
* I think there should be more discussion on the choice of $T_0$. For example, in table 1, why does SGD perform worse when $T_0=0$? It would be good to get a sense of robustness to this hyperparameter.
* Good results on CIFAR. Late-phase weights are shown to boost performance over SGD and to be complementary with SWA. There are some benefits in the OOD setting as well.

### Recommendation / Justification
I vote to accept the paper. The idea is interesting, well-motivated, and seems straightforward to incorporate into existing pipelines. However, the improvements seems modest in some settings (e.g. ImageNet) and for the best performance, it seems like we should still stick to Deep Ensembles.

### Questions
* On the ImageNet experiments, what is the validation accuracy of the pre-trained model?
* Can you comment on the computaional and memory complexity of your algorithm versus vanilla SGD?
* In the comparisons between late phase weights and SGD, do both algorithms consume the same amount of data? If so, this would be good to mention.
* Could the entire network be treated as "late-phase weights"? Would this help performance?

### Minor comments
* I would consider alluding to possible choices of the weight interaction functions $h$ when it is first introduced at the start of 2.1.
* In Algorithm 1: How does the loss function consume three inputs? This is different from when it is initially described.
* It's a bit unclear what is being compared in Figure 2.


(increased score from 6 to 7)

---

> ### Author Response · Authors · 2020-11-19
> **Reply to AnonReviewer1 (1/2)**
>
> Thank you for the thorough and encouraging review. We have carried out additional experiments and reworked the paper following your feedback listed point-to-point below:
>
> * *The weight interaction functions should be more explicitly defined rather than just described in text.*
>
> Thank you for this comment, indeed the BatchNorm and last-layer late-phase weights were missing explicit formulas, which made the presentation in Sect. 2.2 less clear. We have corrected this.
>
> * *I think there should be more discussion on the choice of $T_0$. For example, in table 1, why does SGD perform worse when $T_0=0$? It would be good to get a sense of robustness to this hyperparameter.*
>
> We have extended our $T_0$ sensitivity analysis to CIFAR-10 and searched using a finer step size (Table 12, see also Figure 5); our analysis reveals that this hyperparameter is robust, performance increases as long as it is set to a late-training value. Below, we present a slimmed-down version of Table 12 which shows that for $T_0$ > 80 epochs we improve on top of the baseline in both CIFAR10 (96.16+/-0.12) and CIFAR100 (81.31+/-0.16) in test set accuracy (and out-of-distribution detection, see Figure 6 in the appendix).
>
> +---------+------------------------+---------------------+ \
> |&nbsp;&nbsp;&nbsp;&nbsp;T_0&nbsp;&nbsp;|&nbsp;&nbsp;&nbsp;&nbsp;&nbsp;&nbsp;&nbsp;CIFAR10&nbsp;&nbsp;&nbsp;&nbsp;&nbsp;&nbsp;&nbsp;&nbsp;&nbsp;|&nbsp;&nbsp;&nbsp;&nbsp;&nbsp;&nbsp;CIFAR100&nbsp;&nbsp;&nbsp;&nbsp;&nbsp;|&nbsp;\
> +---------+------------------------+---------------------+\
> |&nbsp;&nbsp;&nbsp;&nbsp;40&nbsp;&nbsp;&nbsp;&nbsp;|&nbsp;&nbsp;&nbsp;96.34&nbsp;+/-&nbsp;0.08&nbsp;&nbsp;&nbsp;|&nbsp;&nbsp;79.69&nbsp;+/-&nbsp;0.11&nbsp;&nbsp;|\
> |&nbsp;&nbsp;&nbsp;&nbsp;80&nbsp;&nbsp;&nbsp;&nbsp;|&nbsp;&nbsp;&nbsp;96.50&nbsp;+/-&nbsp;0.11&nbsp;&nbsp;&nbsp;|&nbsp;&nbsp;81.72&nbsp;+/-&nbsp;0.18&nbsp;&nbsp;|\
> |&nbsp;&nbsp;&nbsp;100&nbsp;&nbsp;&nbsp;|&nbsp;&nbsp;&nbsp;96.45&nbsp;+/-&nbsp;0.08&nbsp;&nbsp;&nbsp;|&nbsp;&nbsp;82.48&nbsp;+/-&nbsp;0.21&nbsp;&nbsp;|\
> |&nbsp;&nbsp;&nbsp;120&nbsp;&nbsp;&nbsp;|&nbsp;&nbsp;&nbsp;96.48&nbsp;+/-&nbsp;0.20&nbsp;&nbsp;&nbsp;|&nbsp;&nbsp;82.87&nbsp;+/-&nbsp;0.18&nbsp;&nbsp;|\
> |&nbsp;&nbsp;&nbsp;140&nbsp;&nbsp;&nbsp;|&nbsp;&nbsp;&nbsp;96.26&nbsp;+/-&nbsp;0.17&nbsp;&nbsp;&nbsp;|&nbsp;&nbsp;82.53&nbsp;+/-&nbsp;0.21&nbsp;&nbsp;|\
> |&nbsp;&nbsp;&nbsp;160&nbsp;&nbsp;&nbsp;|&nbsp;&nbsp;&nbsp;96.23&nbsp;+/-&nbsp;0.11&nbsp;&nbsp;&nbsp;|&nbsp;&nbsp;81.41&nbsp;+/-&nbsp;0.31&nbsp;&nbsp;|\
> +----------+-----------------------+----------------------+
>
> We use the same hyperparameter value on both CIFAR-10 and CIFAR-100 and across the different architectures considered.
>
> * *Good results on CIFAR. Late-phase weights are shown to boost performance over SGD and to be complementary with SWA. There are some benefits in the OOD setting as well.*
>
> Thank you for this appreciation of our results. We have strengthened our CIFAR experiments by considering additional models. On a WRN 28-14 (CIFAR-10), we increase accuracy from 96.75% to 97.45%, on top of SWA, a very high figure for residual networks on this dataset. We obtain an improvement in the order of ~1% on CIFAR-100 with this model.

---

> > ### Author Response · Authors · 2020-11-19
> > **Reply to AnonReviewer1 (2/2)**
> >
> > We would also like to point the reviewer’s attention to our new experiments in the OOD section, with additional new experiments on corrupted data. We have improved the performance of our late-phase averaged model and we now also consider maintaining a late-phase ensemble learned with large $\sigma_0$. This results in significantly improved OOD scores and outperforms deep ensembles (Table 4 and 16 in the paper).
> > \
> > +-------------------------------------------------------------------+--------------------+---------------------+-------------------+\
> > |                       |Testacc.(%)	 |&nbsp;&nbsp; OOD  	 | 	mCE   	|\
> > +-------------------------------------------------------------------+--------------------+---------------------+-------------------+\
> > | Base(SGD)                 |81.35+/-0.16 |0.802+/-0.019 |47.84+/-0.41 |\
> > | Dropout(Mean)(SGD)            |81.31+/-0.20 |0.802+/-0.030 |48.97+/-0.33 |\
> > | Late-phase BatchNorm(SGD)&nbsp;&nbsp;&nbsp;&nbsp;&nbsp;&nbsp;&nbsp;&nbsp;
> >    |82.87+/-0.14 |0.836+/-0.012 |45.59+/-0.25 |\
> > |                       |&nbsp;&nbsp;      |&nbsp;&nbsp;       |&nbsp;&nbsp;      |\
> > | MC-Dropout(SGD)&nbsp;&nbsp;             |81.55+/-0.11 |0.823+/-0.049 |48.09+/-0.36 |\
> > | SWAG(SWA)                &nbsp;|82.12+/-0.03 |0.828+/-0.027 |&nbsp;&nbsp;      |\
> > | BatchEnsemble(SGD)            |81.25+/-0.10 |0.829+/-0.019 |&nbsp;&nbsp;      |\
> > | Late-phase BatchNorm(SGD,non-averaged)&nbsp;&nbsp;&nbsp;&nbsp;&nbsp;&nbsp;|82.71+/-0.10 |0.862+/-0.009 |46.21+/-0.29 |\
> > |                       |&nbsp;&nbsp;      |&nbsp;&nbsp;       |&nbsp;&nbsp;      |\
> > | Deepensemble (SGD)&nbsp;&nbsp;            |84.09&nbsp;    |0.8312&nbsp;    |44.21    	|\
> > | Deepensemble (Late-phaseBatchNorm,SGD)&nbsp;&nbsp;&nbsp;&nbsp;|84.69&nbsp;    |0.8575&nbsp;    |43.15    	|\
> > +-------------------------------------------------------------------+--------------------+---------------------+-------------------+
> >
> > * *On the ImageNet experiments, what is the validation accuracy of the pre-trained model?*
> >
> > We now include this additional column in Table 5; it’s below the SGD baseline, hinting that the optimization of the original model was stopped too early and/or that the restart (reset of the internal state of the optimizer) is beneficial. We also kindly point out our updated ImageNet results (e.g., we increase accuracy from 78.37% to 78.77% on a deeper ResNet-152).
> >
> > * *Can you comment on the computational and memory complexity of your algorithm versus vanilla SGD?*
> >
> > Except when using the hypernetwork weight interaction function (which requires additional tensor products to generate weights), our model results in essentially the same memory and computational complexity of vanilla SGD on one model. We now provide runtimes in Table 17 to make this explicit and emphasize this important feature more strongly in the text.
> >
> > In the comparisons between late phase weights and SGD, do both algorithms consume the same amount of data? If so, this would be good to mention.
> > Yes. Thank you, this important point has been clarified: “All evaluated methods are trained using the same amount of data.” (Section 3.2).
> >
> > * *Could the entire network be treated as "late-phase weights"? Would this help performance?*
> >
> > Thank you for this suggestion. This is a great additional experiment that we ran, maintaining the same data consumption as vanilla SGD. Performance still increases (CIFAR-100 82.17, CIFAR-10 96.32) compared to the SGD baseline (at the expense of the additional memory consumption of a DeepEnsemble), but not as much as when using our proposed late-phase weights (CIFAR-100 82.87, CIFAR-10 96.46). This is possibly explained by the low-dimensionality of our late-phase weight models, which can be efficiently trained using the same data consumption as a single model. We now discuss these new findings in Section 3.2, CIFAR-10 and CIFAR-100 paragraphs.
> >
> > * *In Algorithm 1: How does the loss function consume three inputs? This is different from when it is initially described.*
> >
> > We clarified this notation.
> >
> > * *It's a bit unclear what is being compared in Figure 2.*
> >
> > We reformulated the caption of the figure.
> >
> > Given the additional experiments, baseline comparisons and the new strong performance results we remain open to additional questions and suggestions and welcome any reevaluation of your assessment given you agree that our paper is now much stronger.

---

> > > ### Comment · AnonReviewer1 · 2020-11-22
> > > **Good improvements**
> > >
> > > I’ve read the other reviews and the author’s responses. I think the authors did a thorough job during the rebuttal period and added many experiments/improved clarity in the revision. The resulting paper tells a more compelling story, and I believe should be accepted at ICLR. I have updated my score.
> > >
> > >
> > > >This is a great additional experiment that we ran, maintaining the same data consumption as vanilla SGD. Performance still increases (CIFAR-100 82.17, CIFAR-10 96.32) compared to the SGD baseline (at the expense of the additional memory consumption of a DeepEnsemble), but not as much as when using our proposed late-phase weights (CIFAR-100 82.87, CIFAR-10 96.46).
> > >
> > > Thanks for running this experiment! The results are a bit surprising to me, that using more memory does not equate to better performance.
> > >
> > >
> > > minor nit: “we allow for this increased budget exceptionally for BatchEnsemble” is a bit unclear in the appendix

---

> > > > ### Author Response · Authors · 2020-11-23
> > > > **Thank you**
> > > >
> > > > We are very grateful for your fast reevaluation and happy to read your positive opinion of our work.
> > > >
> > > > > Thanks for running this experiment! The results are a bit surprising to me, that using more memory does not equate to better performance.
> > > >
> > > > A potential explanation is that we trained this large-memory late-phase ensemble with as little data as a single model, to match our late-phase weights setup (so each ensemble member only sees $1/K$ data points). This result emphasizes the importance of selecting a low-dimensional, expressive set of late-phase weights, which do not require (or at least not many) additional passes through the dataset. This will be clarified in the text.
> > > >
> > > > > minor nit: "we allow for this increased budget exceptionally for BatchEnsemble" is a bit unclear in the appendix
> > > >
> > > > We will clarify the Appendix, explaining that we trained the BatchEnsemble exceptionally for 250 epochs (instead of 200), as the hyperparameters provided by the authors were tuned for this larger training duration.

---

### Author Response · Authors · 2020-11-19
**Joint reply to all reviewers**

We thank all four reviewers for their efforts and their constructive feedback. Working these in has substantially improved and strengthened our paper.

Most importantly, the revised version now addresses the most critical point raised by the reviews and highlights the substantial performance increase that our method can achieve.  We now report significantly higher performance increases on larger models, where regularization is essential.

In particular, on a WRN 28-14, we increase accuracy from 96.75% (SWA) to 97.45% (Late-phase+SWA) on CIFAR-10, and from 84.01% (SWA) to 85.00% (Late-phase+SWA) on CIFAR-100. In absolute terms, these are very high accuracies, placing us above all available results for WRNs (without extra training data) in PWC [1] for CIFAR-100; on CIFAR-10 [2] we are seconded only by a result using the deeper WRN 40-10. Furthermore, these results show that our method can be applied complementary to SWA, currently one of the strongest methods for improving generalization in neural networks. On ImageNet, we now increase accuracy from 78.37% to 78.77% on a ResNet-152 (Table 5). This improvement is brought by solely fine-tuning an existing, pretrained model.

In addition, we now compare our method to well-established alternatives that mitigate the computational and memory costs of a deep ensemble: dropout, MC-dropout, and BatchEnsemble. Our method results in stronger performance across all cases considered, including out-of-distribution detection problems, and it is arguably as easy to implement as these. Additionally, we now show that when retaining our late-phase ensemble at the end of training, out-of-distribution scores greatly improve. To facilitate its adoption, we will release to the community a plug-and-play PyTorch module for BatchNorm layers with the final paper.

Finally, we show that our main results can be obtained with a single hyperparameter, the initialization time $T_0$, which is used across CIFAR-10/100. This hyperparameter is robust: the main requirement is to set it to a late-enough value, as shown in our new, finer sensitivity analysis.

To conclude -- in light of the new framing of our results (putting them into perspective to other methods), the additional baselines and several other experiments as well as performance improvements we would kindly ask the reviewers to reevaluate their ratings if they agree that the paper has improved.

In the text below, we answer in detail to the reviewer’s comments and we will remain responsive to any further questions the reviewers may have.

[1] https://paperswithcode.com/sota/image-classification-on-cifar-100 \
[2] https://paperswithcode.com/sota/image-classification-on-cifar-10

---

### Decision · Program_Chairs · 2021-01-07
**Final Decision**

**Decision:**

Accept (Poster)

**Comment:**

This paper proposes to learn an ensemble of weights given a set of base weights from some point late in normal training. The authors apply this approach to a number of configurations and find modest performance improvements for normal test settings and larger improvements for out of distribution settings. While reviewers had some concerns about the size of the improvement relative to baselines, all reviewers agreed that the proposed method is interesting and will likely impact future work, especially given the new experiments provided by the authors. I recommend that the paper be accepted.